# Cytosolic receptors and signaling in antifungal immunity

Sandra Khau [1,2], Guillaume Desoubeaux [1,2,3], Mustapha Si-Tahar [1,2], Elise Biquand [1,2] & Benoit Briard [1,2✉]

## Abstract

**The host innate immune system provides the first line of protection against invading microbial pathogens, including fungi. Recognition of fungi by host pattern-recognition receptors (PRRs) is critical for their clearance. PRRs bind to pathogen-associated molecular patterns (PAMPs) that can be present on the fungal surface, secreted by them, or found in their genetic material, but also damage-associated molecular patterns (DAMPs) released by host cells as a result of fungal infection. These receptors can be located at the cell surface, the endosome, or in the cytosol of host cells. Depending on PRR location and the nature of the molecular patterns (PAMPs/DAMPs) they recognize, their activation induces specific signaling pathways culminating in tailored immune responses. There are two families of innate immune receptors that can principally sense fungi, namely membrane-bound Toll-like receptors (TLRs) and C-type lectin receptors (CLRs). In addition, as phagocytosed fungal pathogens can escape the phagolysosome and reach the cytoplasm, cytosolic sensors such as Nod-like receptors (NLRs), absent in melanoma 2 (AIM2)-like receptors (ALRs), and retinoic acid-inducible gene-I (RIG-I)-like receptors (RLRs) are also important in fungal sensing and play essential roles in antifungal host protection. This review summarizes the cytosolic receptors and the signaling pathways involved in antifungal innate immunity.**

**Keywords** Cytosolic Sensors; Interferon; Inflammasome; Cell Death; Fungi
**Subject Categories** Immunology; Microbiology, Virology & Host Pathogen Interaction; Signal Transduction

## Introduction

Ubiquitous in nature, fungi may be opportunistic pathogens in immunocompromised individuals and patients with underlying health conditions. The biodiversity of fungal species is immense, with an estimated range of 2–11 million species (Phukhamsakda et al, 2022). The number of fungal species known to infect humans is comparatively limited; however, recent estimates suggest that ~6.5 million cases of invasive fungal infections occur globally each year, resulting in around 3.8 million deaths (Denning, 2024; Desoubeaux and Chesnay, 2021). Therefore, fungi have been increasingly recognized as a major healthcare issue in recent years (Bongomin et al, 2017; Brown et al, 2012). The World Health Organization (WHO) recently released a list of 19 fungi representing the greatest threat to human health. Those fungi present an increase in drug-resistance and distribution due to climate change, continuously expanding the at-risk population. Among them, the first critical priority includes three yeast species: *Candida albicans*, *C. auris*, and *Cryptococcus neoformans*; and one filamentous fungus, *Aspergillus fumigatus* (World Health Organization 2022).

Since most human pathogenic fungi exploit weaknesses of the immune system, it is obvious that the immune response of the host plays a crucial role in detecting fungal invaders and facilitating their clearance. Indeed, innate immunity represents the first-line defense, and is mediated by phagocytic cells such as macrophages, neutrophils, and dendritic cells. These cells can sense pathogens, engulf them, and induce an inflammatory response. Several immune signaling pathways have been demonstrated to play pivotal roles in human host defense against fungal pathogens. Mutations affecting the C-type lectin receptor (CLR) pathway, including those in *CLEC7A* (encoding Dectin-1) and *CARD9*, are strongly associated with increased susceptibility to chronic mucocutaneous or vulvovaginal candidiasis as well as fungal meningitis (Lanternier et al, 2015, 2013; Glocker et al, 2009; Ferwerda et al, 2009; Fisher et al, 2017; Geijtenbeek and Gringhuis, 2016; Malamud and Brown, 2024; Dambuza and Brown, 2015; Hatinguais et al, 2023). In parallel, the JAK–STAT signaling axis is critical for orchestrating antifungal immunity, primarily through the regulation of effective Th17 responses. Genetic mutations in STAT1 or STAT3 similarly predispose individuals to recurrent mucocutaneous candidiasis, fungal meningitis, and invasive pulmonary fungal infections (Liu et al, 2011; Duréault et al, 2019; Zheng et al, 2015). Consistent with this, human genetic defects directly impacting the IL-17 pathway confer comparable susceptibility to these fungal pathologies (Puel et al, 2010, 2011; Boisson et al, 2013; Hernández-Santos and Gaffen, 2012; Li et al, 2019). Collectively, these observations highlight the central role of coordinated innate and adaptive immune responses in antifungal host defense.

In order to induce a tailored immune response, pattern-recognition receptors (PRRs) expressed by the immune cells

[1]Inserm, U1100, Centre d'Étude des Pathologies Respiratoires, Tours, France. [2]Université de Tours, Tours, France. [3]Hôpital Universitaire de Tours, Service de Parasitologie-Mycologie-Médecine Tropicale, Tours, France. ✉E-mail: benoit.briard@inserm.fr

recognize different pathogen-associated molecular patterns (PAMPs) expressed by microorganisms, as well as endogenous molecules released from damaged or dying cells known as damage-associated molecular patterns (DAMPs).

Toll-like receptors (TLRs) and C-type lectin receptors (CLRs) are the primary extracellular PRRs expressed at the surface of the host cell membrane. The recognition of extracellular pathogens triggers their phagocytosis by immune surveillance phagocytic cells, which eliminates them and prevents infection. However, phagocytosed fungi can escape the phagolysosome and reach the cytoplasm, or extracellular fungi can still invade host cells with growing hyphae (Sheppard and Filler, 2015; Yang et al, 2014; Fernandes et al, 2018). Thus, a cytosolic detection system for intracellular fungal PAMPs, such as cell wall components, proteases, and nucleic acids, is crucial in controlling the infection. These cytosolic PRRs include the nucleotide-binding oligomerization domain (NOD)-like receptors (NLRs), retinoic acid-inducible gene-I (RIG-I)-like receptors (RLR), and the absent in melanoma 2 (AIM2)-like receptors (ALRs) (Briard et al, 2020b). The cytosolic receptors can sense a large diversity of PAMPs and DAMPs, triggering a robust immune response to the cell invader. The cytosolic sensors can be primarily divided into three functional categories: those mediating interferon (IFN) signaling, those triggering the NF-κB pathway, and those mediating inflammasome activation. These pathways induce activation of transcription factors promoting the expression of proinflammatory genes and activation of cell death pathways such as pyroptosis, apoptosis, and necroptosis (Briard et al, 2021a). Those cell death pathways participate in the immune response to fungal infection.

In this review, we will explore the critical role of cytosolic recognition of fungal pathogens through various cytosolic receptors in driving a robust antifungal immune response. We will exclude intracellular TLR signaling, which is activated within the phagolysosome rather than in the cytoplasm. Mechanisms by which fungal PAMPs access the host cell cytosol to activate these intracellular receptors will be highlighted.

## Cytosolic receptors mediating IFN signaling and host defense against fungal infection

### NOD1 and NOD2

The NOD-like receptors, such as nucleotide-binding oligomerization domain 1 and 2 (NOD1 and NOD2), are intracellular sensors of peptidoglycan, a major component of the bacterial cell wall. Compared to their involvement in activating the inflammatory signaling during bacterial infection, their role in fungal infection has been less extensively studied. For example, the exact mechanisms by which NOD1 and NOD2 interact with other downstream molecular partners and other PRRs remain incompletely understood.

During fungal keratitis, NOD1 and NOD2 were shown to play a role in host defense (Zhang et al, 2014; Xu et al, 2012; Wu et al, 2015). Levels of mRNA and protein expression of NOD1 and its downstream partner receptor-interacting protein 2 (RIP2) were shown to be upregulated when human corneal epithelial cells were infected with heat-inactivated *A. fumigatus* conidia. Increased

NOD1 expression correlated with NF-κB activation and higher levels of IL-6 and TNF cytokine responses (Fig. 1) (Zhang et al, 2014). Similar observations were made for NOD2, not only with corneal epithelial cells (Xu et al, 2012; Wu et al, 2015), but also in A549 lung epithelial cells, THP-1 monocyte (Zhang et al, 2008), and RAW264 macrophage cell lines (Li et al, 2012), where infection with *A. fumigatus* conidia led to increased expression of NOD2 and cytokine release. Altogether, these results suggest a proinflammatory effect of NOD1 and NOD2 in *A. fumigatus* infection. On the other hand, another study showed that the activation of NOD2 after recognizing fungal chitin, a component of the cell wall of most fungal species, mediates an anti-inflammatory IL-10 response (Fig. 1) (Wagener et al, 2014). However, this IL-10 response might be linked to a crosstalk with other PRRs, such as the endosomal receptor TLR9 or mannose receptor (MR), as *Tlr9⁻/⁻* and *Mr⁻/⁻* macrophages exhibit reduced IL-10 production after chitin exposition (Wagener et al, 2014). In this case, the anti-inflammatory response is thought to regulate inflammation and prevent excessive inflammatory damage following fungal infection (Fig. 1).

Other studies have shown that the activation of NOD1 and NOD2 upon *A. fumigatus* challenge can actually inhibit the antifungal response (Gresnigt et al, 2017, 2018b). In the studies of Gresnigt et al, *Nod1⁻/⁻* mice exhibited less fungal outgrowth in the lungs and were protected against invasive aspergillosis compared to wild-type (WT) mice. In vitro experiments with *A. fumigatus* infection of *Nod1⁻/⁻* mice BMDMs resulted in elevated reactive oxygen species (ROS) and cytokine production, as opposed to WT BMDMs (Gresnigt et al, 2017). The same phenomenon was observed in human macrophages and was confirmed by silencing NOD1 using siRNA, resulting in higher fungal killing (Gresnigt et al, 2017). Concomitantly, activating NOD1 with its agonist decreased fungal killing (Gresnigt et al, 2017). This negative regulation of NOD1 on killing capacity can be explained by a reduced dectin-1 expression occurring when NOD1 is activated. Indeed, dectin-1 (*CLEC7A*) is a C-type lectin known to be a crucial PRR in host defense against fungal infections (Fig. 1) (Taylor et al, 2007). Similar observations were made regarding NOD2 (Gresnigt et al, 2018b), as *Nod2⁻/⁻* mice exhibited better resistance than WT mice following *A. fumigatus* conidia challenge. As with NOD1, NOD2 activation was associated with reduced dectin-1 expression. However, nuances of NOD1/NOD2 opposing roles and competition with other signaling pathways exist (Box 1). Overall, despite the differences that can exist between NOD1 and NOD2, their activation seems to be detrimental to the antifungal defense during *A. fumigatus* infection (Fig. 1). Human genetic studies indicate that *NOD2* mutations modulate susceptibility to fungal infections such as invasive aspergillosis by altering cytokine responses and phagocytosis, with some loss-of-function variants paradoxically conferring resistance, underscoring its complex role in antifungal immunity (Gresnigt et al, 2018a). Interestingly, distinct outcomes were observed with *Candida* species. NOD2 was shown to be involved in recognition and induction of cytokine response to *C. parapsilosis* (Patin et al, 2016), but not to *C. albicans* infection (van der Graaf et al, 2006). Thus, these findings suggest that non-*Aspergillus* species need to be further investigated in the context of NOD receptor activation.

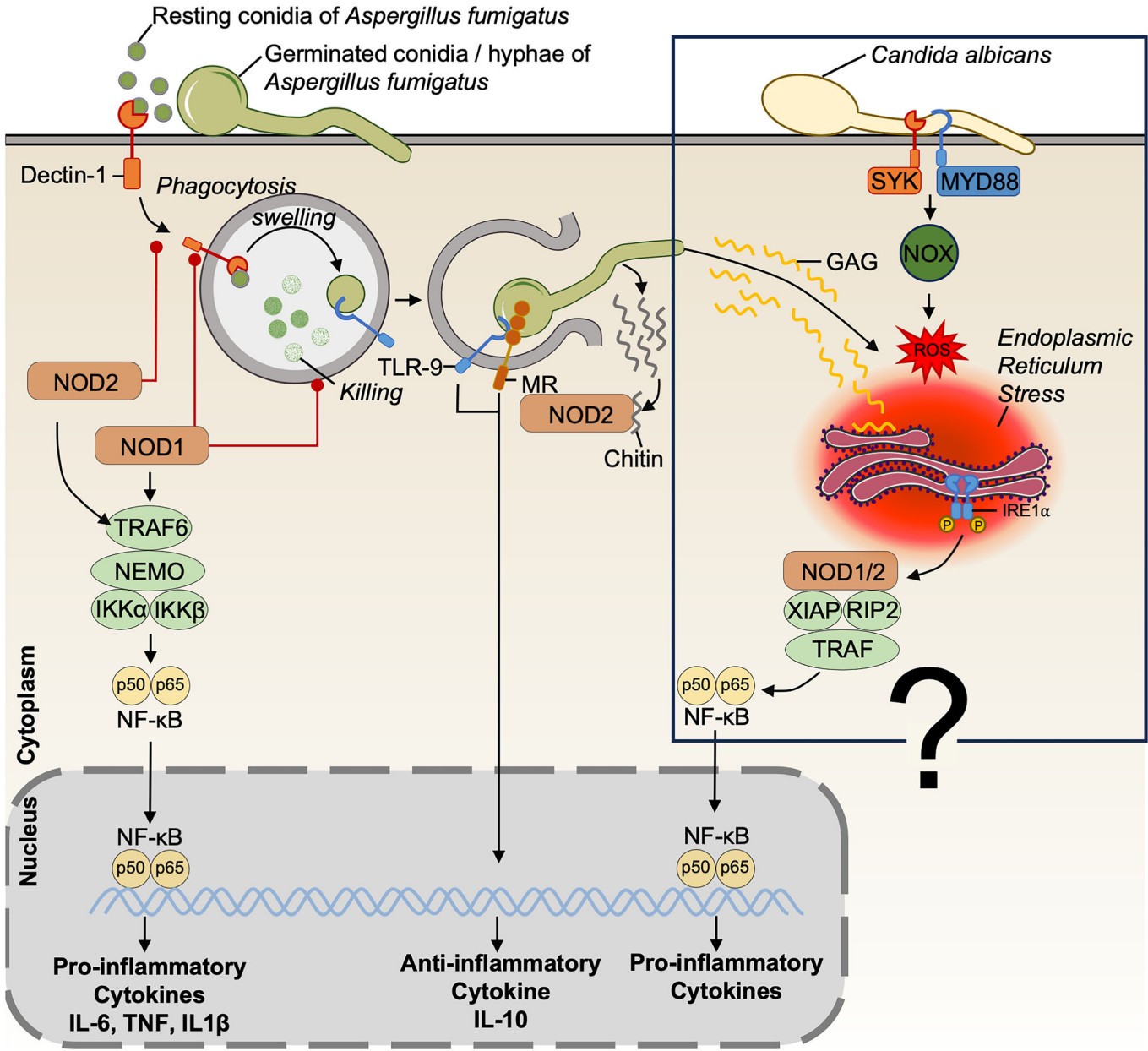

**Figure 1. NOD1 and NOD2 signaling pathways in host defense against fungal infection.**

During *A. fumigatus* and *C. albicans* infections, NOD1 and NOD2 activation triggers the NF-κB pathway and elevates IL-6 and TNF levels in mice. Notably, mice lacking NOD1 or NOD2 exhibit greater resistance to *A. fumigatus* conidia than wild-type controls. In addition, NOD1 and NOD2 exert an inhibitory effect (red dotted lines) on Dectin-1 expression and *A. fumigatus* conidia killing. Furthermore, NOD2-mediated sensing of *A. fumigatus* chitin in conjunction with TLR9 and the Mannose Receptor (MR) induces an anti-inflammatory IL-10 response, thereby fine-tuning the overall immune response. While the specific fungal ligands for NOD1 and NOD2 remain to be confirmed, it is possible that endoplasmic reticulum stress induced by *A. fumigatus* or *C. albicans* infection may stimulate NOD1/2 and their downstream signaling pathways.

Crucially, fungal ligands recognized by NOD1 and NOD2, and the activation mechanism of the receptors remain unclear. Notably, recent evidence suggests that these receptors may detect pathogen-induced endoplasmic reticulum (ER) stress (Kuss-Duerkop and Keestra-Gounder, 2020), a phenomenon observed in *A. fumigatus* (Briard et al, 2020a; Lee et al, 2016, 2020) and *C. albicans* (Awasthi

et al, 2023) infections (Fig. 1). Undoubtedly, investigating ER stress triggered by fungal pathogens and its influence on the NOD pathway will likely yield valuable insights into the fine-tuning of the immune response. More research is needed to fully characterize the complex interactions between these receptors and downstream signaling molecules, as well as the PAMPs they can detect.

> **Box 1    Nuances of NOD1/NOD2 opposing roles and competition with other signaling pathways**
>
> Contrary to Nod1−/− bone marrow-derived macrophages (BMDMs), Nod2−/− BMDMs infected with A. fumigatus showed a reduction in IL-1β and TNF secretion (Gresnigt et al, 2018b). NOD2 also differs from NOD1 in terms of levels of reactive oxygen species (ROS) production (Gresnigt et al, 2018b). Nod2−/− BMDMs did not show any differences in ROS production compared to WT BMDMs, indicating that the heightened fungal killing capacity of Nod2−/− BMDMs is independent of ROS levels and is more likely attributable to an enhanced phagocytic activity (Fig. 1) (Gresnigt et al, 2018b). Indeed, more efficient uptake of conidia was observed with Nod2−/− BMDMs compared to WT BMDMs (Gresnigt et al, 2018b). Those observations imply that NOD1 and NOD2 negatively regulate phagocytosis and killing. The differences in cytokine and ROS production between NOD1 and NOD2 may be due to distinct mechanisms by which these receptors recognize and respond to fungal pathogens. Varying PAMP affinity depending on the receptors, or the sensing of different PAMPs, can occur. Moreover, NOD1 and NOD2 can also compete for downstream molecules, such as RIP2 or CARD9, that are also useful for other pathways important to fight fungal infections, such as Dectin-1 (Taylor et al, 2007; Yang et al, 2011), Dectin-2 (CLEC6A) (Bi et al, 2010), and RIG-I (DDX58) (Jaeger et al, 2015), leading to the induction of a tailored immune response.

# RIG-I-like receptors

Retinoic acid-inducible gene I-like receptors (RIG-I-like receptors or RLRs) are cytosolic receptors that sense RNA and lead to subsequent activation of the mitochondrial antiviral-signaling protein (MAVS) adapter to induce type-I interferon responses, especially against viruses (Seth et al, 2005; Yoneyama et al, 2004; Andrejeva et al, 2004). One of the receptors of the RIG-I-like helicase family, melanoma differentiation-associated protein 5 (MDA5), has been linked to antiviral host defense and was mainly described as a receptor for viral RNA leading to the production of type-I IFNs (IFN-Is) and proinflammatory cytokines (Fig. 2) (Seth et al, 2005; Kato et al, 2006; Andrejeva et al, 2004). Nevertheless, recent studies have shown a broader role of the RLR family receptors, implicating them in antifungal immunity.

Jaeger et al suggested that MDA5 is involved in the immune response to C. albicans infection (Jaeger et al, 2015). Indeed, human macrophages challenged with C. albicans hyphae, but not with its yeast form, induce strong expression of genes that are crucial components of the RLR signaling pathway, such as IFIH1, ISG15, IL8, and TRIM25 (Fig. 2) (Jaeger et al, 2015). Notably, IFIH1 (Interferon-induced helicase C domain-containing protein 1) encodes MDA5, the receptor for long double-stranded (ds)RNA (Rehwinkel and Gack, 2020), whereas ISG15 and TRIM25 play a role in the RIG-I signaling cascade that is triggered by shorter dsRNA (Jaeger et al, 2015). Those two pathways converge at the signaling adapter MAVS (Fig. 2) (Biondo et al, 2011; Pekmezovic et al, 2022; Dutta et al, 2020; Espinosa et al, 2017). In contrast, downregulation of IFIH1 expression is associated with increased susceptibility to candidiasis, as observed in patients carrying IFIH1 gene polymorphisms compared to healthy individuals. IFIH1 mutations could impair MDA5 function, affecting RNA binding and antifungal responses. Infection of MDA5-deficient splenocytes or PBMCs with IFIH1 mutations showed altered pro- and anti-inflammatory cytokine profiles compared to WT controls, potentially contributing to increased susceptibility to disseminated candidiasis (Jaeger et al, 2015). Models of C. albicans infection put in

contact with PBMCs from healthy donors upregulated IFIH1, thus providing further evidence for its actual role in host defense.

However, MDA5 activation during C. albicans infection might also be harmful in certain conditions (Chen et al, 2024), similarly to type-I IFN (Majer et al, 2012b) (Fig. 2). Surprisingly, the deleterious effect of MDA5 seems to be independent of IFN production, and is instead driven by heightened macrophage sensitivity to cell death, which leads to liver failure (Chen et al, 2024). Such discrepancies between studies investigating the role of MDA5 during candidemia are likely due to the differences in models used (Chen et al, 2024; Wang et al, 2020). MDA5 appears to be crucial during severe and acute infection, when an efficient IFN response is required. In contrast, in a less virulent infection model, MDA5 becomes detrimental due to its role in promoting apoptosis.

Regarding pulmonary aspergillosis (Wang et al, 2020), MDA5 was also shown to be necessary for host resistance against the pathogen. The critical role of the MDA5/MAVS pathway was confirmed by the identification of IFIH1 and MAVS polymorphisms, which are associated with an increased incidence of invasive infection in hematopoietic stem-cell transplantation (HSCT) recipients (Wang et al, 2022). Indeed, patients carrying IFIH1 mutations show altered pro- and anti-inflammatory cytokine profiles, correlating with increased infection risk (Jaeger et al, 2015). In vitro, transfection of A. fumigatus RNA into murine fibroblasts induced secretion of IFN-α and CXCL10, suggesting that the host cells could sense A. fumigatus RNA intracellularly and mount an IFN-dependent response. This type-I IFN is partially dependent on the MDA5 pathway, given that in MDA5-deficient (Ifih1−/−) fibroblasts, IFN-α levels were only moderately reduced. This result implies that another dsRNA receptor could be involved in this IFN-α response, probably RIG-I or a member of DEAD-box or DEAH-box families (DDX and DDHX, respectively). Noteworthy, to date RIG-I helicase stricto sensu has surprisingly not been shown to play a role in fungal infections (Wang et al, 2020). However, the IFN response seems to totally depend on the RLR pathway, as MAVS-deficient cells showed a complete decrease in IFN gene expression, especially the type-III IFNs, which are crucial in response to A. fumigatus (Fig. 2) (Wang et al, 2020; Espinosa et al, 2017). In vivo, MDA5- or MAVS-deficient mice were more susceptible to A. fumigatus infection than WT mice (Wang et al, 2020): when infected with A. fumigatus, they presented fewer macrophages and neutrophils in their airways than WT mice (Wang et al, 2020), while these cells are acknowledged to play a critical role in clearing A. fumigatus from the lungs (Schaffner et al, 1982; Brown, 2011). In the same line, IFIH1-deficient mice showed less clearance of conidia compared to the WT mice (Wang et al, 2020). Moreover, MDA5/MAVS activation was shown to be required for the antifungal activity of neutrophils, as a higher number of live conidia was present in Ifih1−/− mice neutrophils compared to WT mice neutrophils (Wang et al, 2020). The same observations in human and mouse alveolar macrophages support this critical role of the MDA5/MAVS pathway (Wang et al, 2022).

Although it is clear that the MDA5/MAVS pathway plays a crucial role in the host immune response against fungal infections, the specific fungal ligands that activate MDA5 remain unclear. Notably, the size of RNA appears to be a critical determinant, as MDA5 preferentially recognizes longer double-stranded RNA molecules, distinguishing its role from other sensors such as RIG-I. For a detailed discussion of the RNA ligands, the putative role of mycoviruses, and their mechanistic implications, see Box 2. Further investigations are needed to clarify the mechanisms underlying MDA5 activation during fungal infections.

Living *A. fumigatus* conidia can trigger MDA5/MAVS activation, while heat-killed conidia fail to induce this response, likely due to RNA degradation or structural alterations (Fig. 2) (Wang et al, 2020). Notably, total RNA extracts from *A. fumigatus* can activate the MDA5 pathway, confirming that RNA is the primary ligand (Wang et al, 2020). A possible explanation is that MDA5 may detect long dsRNA not produced by the fungus itself but by mycoviruses infecting the cytoplasm of pathogenic fungal strains (Kanhayuwa et al, 2015; Rocha et al, 2025). This mechanism could explain why fungi preferentially activate MDA5 over RIG-I. This hypothesis is supported by the fact that dsRNA length is critical for RLR specificity: RIG-I detects shorter dsRNAs (<500 bp), whereas MDA5 recognizes longer dsRNAs (up to 1 kbp) (Im et al, 2023; Reikine et al, 2014). In that respect, *A. fumigatus* tetramycovirus-1 (AfuTmV-1) produces long dsRNAs ranging from 1 to 2.5 kbp, which may therefore exhibit higher affinity for MDA5, thereby potentially driving its activation (Fig. 2) (Kanhayuwa et al, 2015). During *Candida* infection, hyphal forms appear to be required for MDA5 activation, in contrast to yeast stages. This preferential recognition may result from the initial detection by cell surface receptors, such as TLRs and CLRs, followed by internalization and phagolysosomal escape. However, it is possible that *C. albicans* might also harbor a mycovirus (Mehta et al, 1982; Sharma et al, 2011).

## cGAS–STING

The activation of cyclic GMP-AMP synthase (cGAS) and its downstream signaling partner, the stimulator of interferon genes (STING), is essential for detecting the presence of cytosolic double-stranded (ds)DNA or RNA:DNA hybrids from microbial pathogens or self-DNA (Mankan et al, 2014; Tan et al, 2018). Mechanistically, detection of abnormal cytosolic DNA activates cGAS, resulting in the production of the secondary messenger cyclic GMP-AMP (cGAMP). cGAMP then binds to STING, triggering downstream signaling through TBK1 and IRF3 (Fig. 3) (Briard et al, 2020b). While the role of cGAS–STING signaling in antiviral and anti-bacterial immunity is well established, its function during fungal infection remains poorly understood.

Recent studies have shown that the cGAS–STING pathway negatively influences the immune response during *C. albicans* infection (Brown Harding et al, 2024; Chen et al, 2023a). Interestingly, *Candida* DNA could be delivered into host cells *via* fungal extracellular vesicles acting as carriers (Fig. 3) (Brown Harding et al, 2024). Then, upon sensing of *C. albicans*, cGAS–STING activation triggers type-I IFN signaling, contributing to immunopathology and leading to fatal outcomes in mice (Majer et al, 2012a; Brown Harding et al, 2024). Likewise, human polymorphisms in the *MB21D1* (encoding cGAS) and *TMEM173* (encoding STING) genes have been associated with altered production of proinflammatory cytokines, including TNF and IL-6 (Brown Harding et al, 2024). However, no specific polymorphisms have been identified to date in patients with higher susceptibility to invasive fungal infections.

Furthermore, STING is also known to have non-IFN-related functions. Upon activation by cGAMP, STING translocates to the endoplasmic reticulum (ER), where it facilitates light-chain 3 (LC3) recruitment for phagolysosome formation (Fig. 3) (Gui et al, 2019; Liu et al, 2019). Specifically during *C. albicans* infection, STING follows a similar pathway, translocating to the ER and then to the phagosome. There, STING directly interacts with cytoplasmic tyrosine-protein kinase (Src) through its N-terminal 18 amino acids, preventing Src-mediated phosphorylation of SYK (Fig. 3). This inhibition thereafter blocks SYK signaling, thereby impairing an efficient antifungal immune response (Chen et al, 2023a).

The role of the cGAS–STING pathway during *A. fumigatus* infection remains poorly understood. First, one could question the role of extracellular vesicles during aspergillosis, since, like *C. albicans*, *Aspergillus* is also capable of producing them (Freitas et al, 2023; Rizzo et al, 2020). A recent study demonstrated that *A. fumigatus* keratitis activates the cGAS–STING pathway, as evidenced by increased expression of cGAS, STING, and TBK1 in both human corneal epithelial cells and mouse corneas (Fig. 3) (Han et al, 2021). Inhibition of cGAS with the RU.521 compound reduces the severity of keratitis in mouse corneas, highlighting the detrimental role of the cGAS–STING pathway in this infection model, like in candidiasis. Conversely, during pulmonary aspergillosis, cGAS–STING pathway activation appears protective, as its inhibition exacerbates pulmonary damage (Peng et al, 2023). This opposing effect is probably due to differences in the immunological response of the two models. In the case of fungal keratitis, infection-induced type-I IFN signaling and inflammation were previously shown to contribute to detrimental corneal ulceration during fungal proliferation (Zhong et al, 2018). In contrast, mice deficient in IFNAR1 exhibit heightened susceptibility to systemic *A. fumigatus* infection following either intratracheal instillation or injectable challenge (Ramirez-Ortiz et al, 2011; Espinosa et al, 2017). Altogether, these observations suggest a pivotal role for the cGAS–STING pathway in regulating interferon responses to *A. fumigatus* infection (Fig. 3). However, research has so far primarily relied on gene downregulation or pharmacological inhibition to assess the role of the cGAS–STING pathway in the context of aspergillosis. Therefore, further studies employing more comprehensive approaches, such as genetic models for in vivo investigation, are needed to elucidate the implication of the cGAS–STING pathway.

## Inflammasome receptors in immune signaling and host defense against fungal pathogens

Inflammasomes are stimulus-induced cytoplasmic multiprotein complexes. They typically consist of a sensor, such as NLRP3, AIM2, NLRC4, NLRP1, or Pyrin, along with the adapter protein ASC (apoptosis-associated speck-like protein containing a caspase activation and recruitment domain (CARD)), which acts as a bridge to the effector protein caspase-1 (Christgen et al, 2020). Usually, the inflammasome sensors have a pyrin domain (PYD) and also contain a leucine-rich-repeat domain (LRR) for NLRP3 and NLRC4, HIN-200 domain for AIM2, a nucleotide-binding domain (NBD) for NLRP1, or a B30.2 domain for Pyrin. These domains sense the stimulus and initiate assembly with ASC. The ASC protein has a PYD and a CARD domain. Upon sensor activation, the interaction between the PYD domains of the sensor and ASC facilitates recruitment of the protein pro-caspase-1 through the CARD-CARD interaction (except for NLRC4, which already includes a CARD), leading to caspase-1 activation and formation of a fully active inflammasome complex. Activated caspase-1 cleaves the protein gasdermin D (GSDMD), releasing the lytic N-terminal domain that forms pores in the plasma membrane, inducing pyroptosis, an inflammatory form of programmed cell death. Simultaneously, the proinflammatory cytokines pro-IL1-β and pro-IL-18 are cleaved, and their active biological forms are released through the GSDMD pores during pyroptosis. These inflammasome mechanisms have been characterized as critical mediators of fungal sensing and drivers of

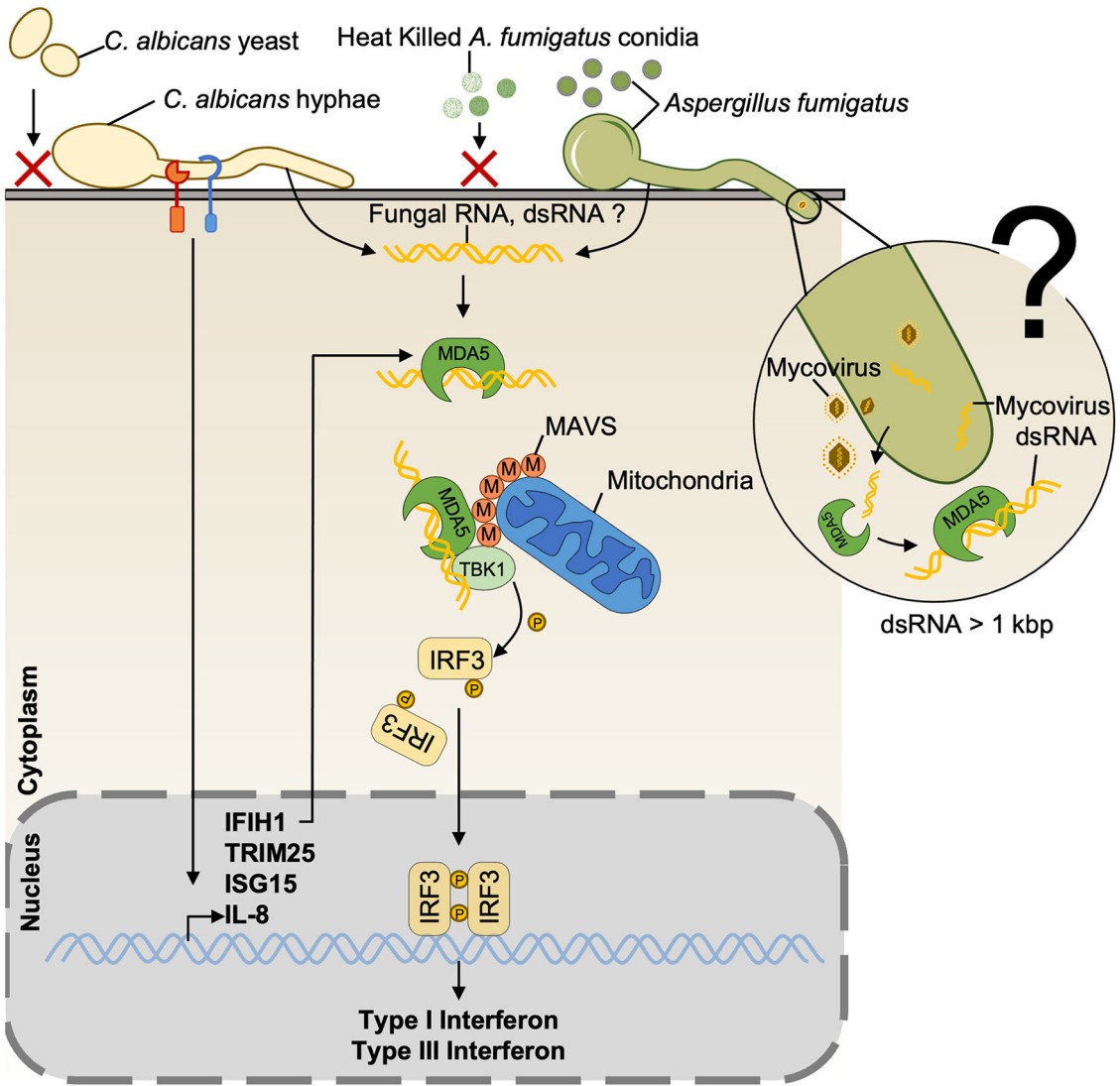

**Figure 2. MDA5 signaling pathway activation by fungal infection.**

Fungal RNA is detected by the cytosolic receptor MDA5, which activates the mitochondrial antiviral-signaling protein (MAVS) and promotes type-I and III interferon responses. Sensing of fungal hyphae by Dectin-1 and Toll-like receptors (TLRs) further elevates MDA5 expression, enhancing pathogen recognition. Besides, it is possible that mycoviruses infecting fungi might also activate MDA5 through sensing of their dsRNA. In *C. albicans* infection, the deleterious effects of type-I interferon are evidenced by reduced susceptibility in *Ifih1*[-/-] and *Ifnar1*[-/-] mice. In contrast, during *A. fumigatus* infection, the protective effects of type-I interferon are demonstrated by increased susceptibility in *Ifih1*[-/-] and *Ifnar1*[-/-] mice.

proinflammatory responses, and are critical in developing an effective Th1 response against fungi (Ketelut-Carneiro et al, 2015; Kurita et al, 2005; Stuyt et al, 2002; Mencacci et al, 2000).

## NLRP3 inflammasome

### Canonical NLRP3-ASC-Caspase-1 inflammasome

Among the different inflammasomes, the NLRP3 is the main one associated with fungal infection. Two signals are required for the canonical NLRP3 inflammasome assembly and activation. In the context of fungal infection, the first signal (priming signal) is achieved via TLR recognition of fungal PAMPs (Mambula et al,

2002; Meier et al, 2003; Netea et al, 2003b; Wang et al, 2001; Ramaprakash et al, 2009; Ramirez-Ortiz et al, 2008) or CLR (Hohl et al, 2005; Steele et al, 2005; Barrett et al, 2009; Hise et al, 2009; Gross et al, 2009) and the downstream NF-κB signal that induces synthesis of pro-IL1-β cytokine and expression of other proinflammatory genes (Briard et al, 2021a). The second signal (activation signal) triggers the assembly of the inflammasome with an active caspase-1 to cleave the pro-IL-1β and pro-IL-18 cytokines into their active form. This two-step mechanism ensures the activation of the inflammasome and pyroptosis only if the immune cells are stimulated by a PAMP or DAMP deemed as dangerous for the host. Historically, NLRP3 inflammasome activation was described with cell wall-purified components as the zymosan, mannan, and curdlan (Lamkanfi et al, 2009; Kumar et al, 2009),

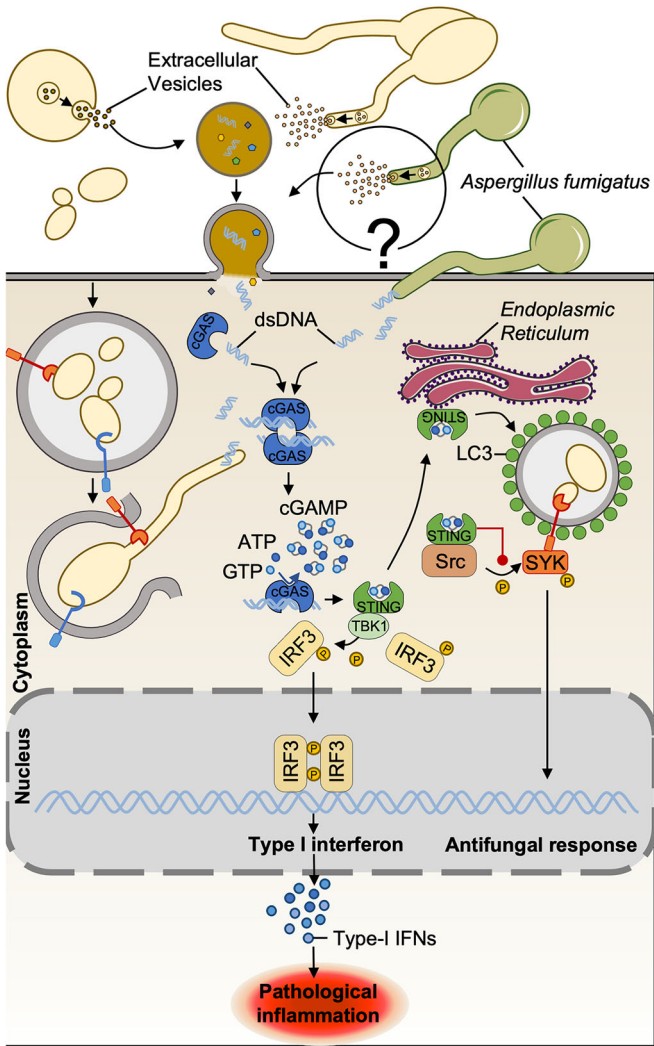

**Figure 3.  Sensing of double-stranded DNA (dsDNA) by the cGAS–STING pathway during fungal infection.**

Upon fungal infection, cytosolic dsDNA is detected by cGAS, leading to STING activation and subsequent production of type-I interferons. Activated STING then translocates to the endoplasmic reticulum, where it promotes recruitment of light-chain 3 (LC3) to facilitate phagolysosome formation. STING interacts with Src, thereby inhibiting SYK activation and subsequent antifungal signaling. Recently, *C. albicans* was shown to produce extracellular vesicles containing dsDNA that is sensed by cGAS, raising the possibility that a similar mechanism might occur during *A. fumigatus* infection. The detrimental role of the cGAS–STING pathway in *A. fumigatus* and *C. albicans* infection models is demonstrated by the reduced susceptibility of *cGas*[-/-] or *Sting*[-/-] mice.

with the fungal β-glucan and mannan being mainly recognized by the TLR2/MyD88/TRIF pathways (Lamkanfi et al, 2009).

During infection of macrophages by *C. albicans*, pro-IL-1β transcription is regulated through both TLR2/MyD88 and CLR/Dectin-1/SYK pathways, activating the downstream CARD9/BCL10/MALT1 complex that is essential for IL-1β secretion (Fig. 4) (Hise et al, 2009). In contrast, during infection of dendritic cells by *C. albicans*, the priming signal only requires the CLR/Dectin-1/SYK pathway (Gross et al, 2009). The observed variation may arise from differences in receptor expression and functionality between the

two cell types. During macrophage infection by *A. fumigatus*, simultaneous engagement of both CLRs/Dectin-1 or 2/SYK pathway and TLRs (-2, -3, -4, and 9)/MyD88 pathway is required to mediate the priming of inflammasome activation (Fig. 4) (Briard et al, 2019; Karki et al, 2015; Saïd-Sadier et al, 2010).

Concerning the second signal for the assembly and activation of the NLRP3 inflammasome, fungal infections induce ROS and phagosomal rupture with potassium (K⁺) efflux, which are recognized as cellular disturbances by the NLRP3 sensor and, in turn, activate caspase-1 and IL-1β cleavage (Fig. 4) (Saïd-Sadier et al, 2010; Gresnigt and van de Veerdonk, 2014; Pietrella et al, 2013; Guo et al, 2014; Tavares et al, 2015). However, the mechanisms by which fungal PAMPs generate this second signal necessary for robust inflammasome oligomerization are complex, relying on multiple PAMPs and varying depending on the pathogen. For example, the secreted aspartic proteases (Saps), Sap2 and Sap6, from *C. albicans* were the first fungal proteins shown to initiate a canonical NLRP3 inflammasome activation signal and induce IL-1β release in human monocytes (Fig. 4) (Pietrella et al, 2013). Subsequently, *Candida* ergosterol or cell wall components, e.g., highly mannosylated proteins or β-(1,3)-glucan, were also evidenced to elicit inflammasome activation (Fig. 4) (Koselny et al, 2018; O'Meara et al, 2015). However, the precise molecular mechanisms activated when the polysaccharide is present in the cytosol, which is sometimes observed with β-(1,3)-glucan in macrophages (Briard et al, 2019), remain unknown.

For the activation of NLRP3–caspase-1 inflammasome during *Candida* infection, the transition to hyphal growth with rearrangement of the fungal cell wall sounds critical (Kasper et al, 2018). Some studies have identified candidalysin, a cytolytic pore-forming toxin secreted by the hyphal form of *C. albicans*, as a key driver of inflammasome activation (Kasper et al, 2018; Rogiers et al, 2019). Candidalysin plays a pivotal role in facilitating hyphal escape by acting within the same pathway as GSDMD (Olivier et al, 2022). Moreover, macrophage permeabilization *via* GSDMD pore formation, hyphal escape, and IL-1β release, appear reduced, when using mutant strains that do not produce candidalysin (*ece1Δ/Δ*) (Fig. 4) (Kasper et al, 2018; Ding et al, 2021). Recently, another work demonstrated that loss of GSDMD, but not NLRP3, reduces hyphal escape from macrophages (Olivier et al, 2022). In this context, trapping fungi inside deficient macrophages reduced inflammatory activation, but did not increase fungal killing. *Gsdmd*[-/-] mice were found to be less susceptible to systemic *C. albicans* infection (Ding et al, 2021), suggesting that inhibition of hyphal escape dampens inflammation rather than directly increases pathogen killing. Therefore, one could assume that fungal containment may be beneficial to clear fungi and avoid damage induced by hyphal growing, thus reducing immunopathology. Nonetheless, activation can still occur with phagosome-contained *C. albicans* (O'Meara et al, 2018), demonstrating that filamentation alone is not sufficient for inflammasome activation to drive IL-1β release and pyroptosis. Mutant strains that do not form hyphae are still able to activate pyroptosis (O'Meara et al, 2015; Wellington et al, 2013).

In addition to the candidalysin-forming pores or mechanical escape by hyphae, competition for glucose can also induce cell death. Indeed, hyphal growth consumes glucose from the host cell, and triggers starvation in macrophages, providing a cellular disturbance signal for inflammasome activation (Tucey et al, 2020). However, in such a context, NLRP3 inflammasome activation levels remain low (Weerasinghe et al, 2023).

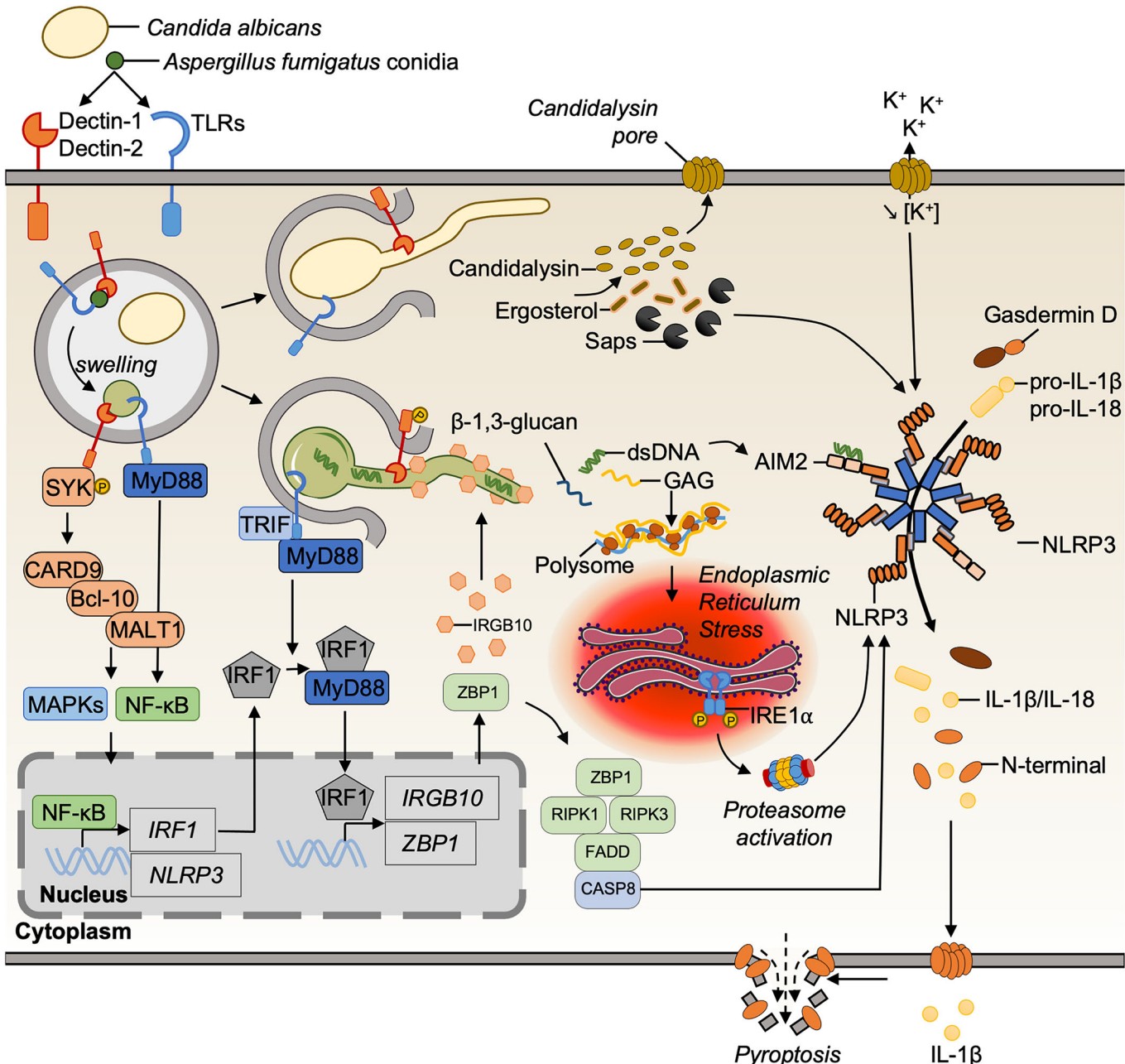

**Figure 4. Inflammasome activation by cytosolic sensing of fungal infection.**

*Candida albicans* and *Aspergillus fumigatus* infect myeloid cells and activate various inflammasome receptors. Fungal pathogens engage C-type lectin (Dectin-1 and -2) and Toll-like receptor pathways that initiate priming and upregulate key inflammasome components. In *C. albicans*, the second activating signal arises from candidalysin, secreted aspartic proteases (SAPs), or ergosterol, each of which can induce potassium efflux and drive NLRP3 inflammasome activation. During *A. fumigatus* infection, the IRGB10 protein targets the fungal surface, facilitating the release of ligands, including β-1,3-glucan, dsDNA, and galactosaminogalactan (GAG), which activate both NLRP3 and AIM2 inflammasomes. GAG also induces translation inhibition and endoplasmic reticulum stress, further contributing to NLRP3 activation. The Z-DNA cytosolic sensor ZBP1 similarly regulates inflammasome activity in response to *C. albicans* and *A. fumigatus*. Once activated, inflammasomes process GSDMD, pro-IL-1β, and pro-IL-18 leading to pyroptosis and inflammation. The protective role of inflammasome signaling in *A. fumigatus* infection is highlighted by the increased susceptibility of $Nlrp3^{-/-}$, $Casp1^{-/-}Casp11^{-/-}$, $Casp1^{Null}$, and $Casp11^{-/-}$ mice.

As with *Candida* species, the morphotype of *A. fumigatus* plays a crucial role in inflammasome activation. It has been reported that hyphae of *A. fumigatus*, but not conidia, trigger NLRP3 activation through ROS production and K+ efflux (Saïd-Sadier et al, 2010) (Fig. 4). Recently, the polysaccharide

galactosaminogalactan (GAG) was identified as a driver of NLRP3 inflammasome assembly. This carbohydrate is present at the surface of germ tubes and hyphae of the *A. fumigatus* cell wall, but absent from resting conidia. When present, GAG acts as a PAMP by interacting with ribosomes and polysomes of host cells (Briard

et al, 2020a). This interaction leads to transcriptional inhibition of the infected cells and the accumulation of unfolded proteins, inducing endoplasmic reticulum stress that is sensed as a danger signal to trigger NLRP3 activation (Briard et al, 2020a). As a consequence, overproduction of GAG by *A. fumigatus* enhances inflammasome activation and elicits a protective response, with a better survival rate for mice compared to mice infected with a strain deficient in GAG production (Briard et al, 2020a). However, some studies have shown that GAG reduces IL-1β secretion (Speth et al, 2019; Henriet et al, 2016). Indeed, cell wall-associated GAG can mask β-1,3-glucan and other PAMPs on hyphae, reducing recognition by innate receptors, and GAG-deficient hyphae elicit more proinflammatory cytokines from dendritic cells (Gravelat et al, 2013). In human peripheral blood mononuclear cells (PBMCs) from chronic granulomatous disease (CGD) patients, *Aspergillus nidulans* strains unable to produce GAG were more virulent and induced higher levels of IL-1β secretion than *A. fumigatus* (Henriet et al, 2016; Speth et al, 2019). Discrepancies in IL-1β secretion induced by GAG may result from differences between immune cell types, species-specific responses in human *versus* mouse systems, variations in mouse models used for in vivo studies, or the use of different *Aspergillus* strains or morphotypes.

Indeed, the form, localization, and acetylation state of GAG shape host inflammatory responses. The cytosol-localized, deacetylated GAG acts as a PAMP that activates NLRP3 and drives IL-1β production. On the other hand, cell wall or extracellular GAG can induce IL-1Ra release in human PBMCs, dampening IL-1-mediated inflammation (Gresnigt et al, 2014).

In terms of human genetic evidence, several polymorphisms in the *Nlrp3* gene have been shown to influence susceptibility and severity of fungal infections by modulating inflammasome activation and IL-1β production. Gain-of-function variants of *Nlrp3* can result in heightened inflammasome activation, contributing to both protective immunity and excessive inflammation, highlighting the dual role of NLRP3 in antifungal defense and inflammation (Verma et al, 2012; Zhong et al, 2022b).

Overall, canonical NLRP3 activation involves caspase-1 and seems to be essential for mediating the IL-1β inflammasome-dependent response (Lamkanfi et al, 2009; Saïd-Sadier et al, 2010; Pietrella et al, 2013; Tavares et al, 2015). However, it was shown to be dispensable in human dendritic cells for IL-1β secretion (Gringhuis et al, 2012), which can possibly be explained by activation of other inflammasome type(s).

# Non-canonical NLRP3 inflammasomes

## Caspase-11

In addition to the NLRP3-dependent inflammasome, some fungal infections can induce a non-canonical NLRP3 inflammasome that is caspase-11-dependent. So far, caspase-11 was mostly described for its activation in response to lipopolysaccharide (LPS) (Shi et al, 2014; Hagar et al, 2013) and oxidized phospholipids (Zanoni et al, 2016). Mechanistically, caspase-11 cooperates with caspase-1 activation and plays a critical role in mediating IL-1β secretion (Gabrielli et al, 2015), as no caspase-1 and IL-1β active forms were detected in *casp-11*$^{-/-}$ mice, while less IL-1β was secreted in *casp-11*$^{-/-}$ neutrophils (Sun et al, 2018). More recently, *casp-11*$^{-/-}$ mice were found to be more susceptible to *A. fumigatus* lung or eye infection compared to WT mice (Man et al, 2017; Sun et al, 2018) and exhibited more hyphal mass growing (Sun et al, 2018). However, caspase-11 in BMDCs seems to have no role in inflammasome activation (Man et al, 2017). In another study, Sap2 and Sap6 of *C. albicans*, in addition to their implication in canonical NLRP3 inflammasome activation, were shown to activate caspase-11 through type-I IFN production in murine macrophages (Gabrielli et al, 2015). In other models, results are sometimes discordant, since during *Paracoccidioides brasiliensis* infection of BMDMs and mice, caspase-11 deficiency did not impair IL-1β secretion (Ketelut-Carneiro et al, 2019). However, its activation was shown to promote faster pore formation and subsequent pyroptosis in BMDMs, which participate in the release of IL-1α, enhancing effector functions of innate cells and human cell responses (Ketelut-Carneiro et al, 2019). Therefore, one could assume that the caspase-11 action to control fungal infection may be specific to the cell type and the kind of infection.

Overall, the activation of a caspase-11-dependent non-canonical inflammasome can constitute an activator and regulator of fungal infection-triggered inflammatory response. However, deeper studies are required to determine how caspase-11 is activated and whether its activation depends on specific fungal antigens.

## Caspase-8

In addition to the NLRP3-dependent inflammasome, some fungal infections can induce NLR-independent, caspase-8-dependent inflammasome activation. Caspase-8 is well-known for its important role in apoptosis (Boldin et al, 1996; Kumar, 2007). It can regulate IL-1β secretion through inflammasome activation-dependent or independent mechanisms (Gurung et al, 2014; Gurung and Kanneganti, 2015). In the former case, caspase-8 acts by mediating gene transcription downstream of Dectin-1/SYK pathway and CR3, another receptor implicated in β-glucan sensing, through the formation of a complex with CARD9/BCL10/MALT1, activating NF-κB. The other inflammasome-dependent function of caspase-8 involves its presence directly in the NLRP3-ASC-caspase-1 scaffold, and its action in concert with caspase-1 (Karki et al, 2015) in the processing of pro-IL-1β into bioactive IL-1β (Gringhuis et al, 2012; Ganesan et al, 2014). In *Cryptococcus neoformans* infection, a non-canonical caspase-8 inflammasome can also be activated, inducing IL-1β secretion and host cell death, especially in dendritic cells deficient for caspase-1 (Chen et al, 2015). Regarding its inflammasome-independent action, caspase-8 was shown in human dendritic cells upon *C. albicans* infection to directly cleave IL-1β, without using caspase-1 (Gringhuis et al, 2012). Likewise, infection of bone marrow-derived dendritic cells (BMDCs) led to the processing of pro-IL-1β upon *P. brasiliensis* recognition, not only by caspase-1-dependent inflammasome responses, but also by caspase-8-dependent mechanisms. Moreover, the process was enhanced in the absence of caspase-1 (Ketelut-Carneiro et al, 2018), which may be explained by competition between caspase-8 and caspase-1 in order to bind to ASC protein.

Hence, caspase-8 can play diverse roles and engage multiple proteins depending on the stimulus. More investigations are required to fully elucidate the molecular mechanism and the nature of the different activators leading to their varying actions during fungal challenge.

## AIM2

The AIM2 inflammasome is known to recognize cytosolic dsDNA (Hornung et al, 2009). While during bacterial infection, DNA is delivered to AIM2 with the assistance of interferon-inducible GTPases (Xue et al, 2019), the role of AIM2 inflammasome in response to fungi is less clear. Interestingly, the interferon-inducible GTPase IRGB10 is required for inflammasome activation during *A. fumigatus* infection (Briard et al, 2019), which may participate in the dsDNA release (Fig. 4). Surprisingly, IRGB10 activity was independent of other interferon-inducible GTPases, such as guanylate-binding proteins (GBPs), whereas during bacterial infection, a specific sequence of GBP recruitment is essential for targeting cytosolic bacteria, licensing IRGB10 recruitment, bacterial lysis, and dsDNA release to activate AIM2 inflammasome (Meunier et al, 2015; Kutsch et al, 2020; Man et al, 2015, 2016). It will be interesting to explore the role of other interferon-inducible GTPases to better define their contribution and the mechanisms governing their recruitment during fungal infection.

The cooperative activation of NLRP3 and AIM2 inflammasomes helps in forming a single cytoplasmic inflammasome platform with the ASC, and both caspase-1 and -8, mediating an efficient response (Fig. 4). Indeed, mice lacking both AIM2 and NLRP3 were highly susceptible to infection and failed to control hyphal growth compared to either WT animals or rodents lacking a single inflammasome (Karki et al, 2015). Interestingly, the surface of *A. fumigatus* (Rajendran et al, 2013) and *C. albicans* (Martins et al, 2010) biofilms is covered with extracellular DNA, which may explain the activation of AIM2 in response to infection.

Interestingly, a negative effect of AIM2 was recently identified during yeast infection. *Aim2$^{-/-}$* mice infected with *C. albicans* had a better survival rate and attenuated liver damage, with lower levels of apoptosis (Jiang et al, 2024). These findings highlight the intriguing dual role of AIM2, functioning both within the inflammasome and through inflammasome-independent mechanisms. Indeed, it is well established that type-I IFNs, known to have detrimental effects, regulate the expression of GTPases, potentially explaining the negative impact of AIM2 in *Candida* infections. Further investigation into interferon-inducible GTPase function and interplay with AIM2 will yield further needed insights into the underlying mechanisms.

## NLRC4

During in vitro *C. albicans* infection, it was reported that the NLRC4 inflammasome is dispensable for IL-1β production in macrophages and dendritic cells, as *Nlrc4$^{-/-}$* BMDCs showed no difference in IL-1β secretion compared to WT BMDCs (Gross et al, 2009). However, in vivo, it has been shown that the NLRC4 inflammasome can be activated upon *C. albicans* infection, and it is necessary for IL-1β and IL-18 production, playing an important role in immune resistance to oral candidiasis (Fig. 5) (Tomalka et al, 2011). Conversely, in vaginal candidiasis, NLRP3 and NLRC4 were non-redundantly activated. Indeed, *Nlrc4$^{-/-}$* mice developed severe disease symptoms and inflammatory responses concomitant with NLRP3 activation. In this case, NLRC4 negatively regulates NLRP3 activation in epithelial cells and in a murine model of vaginal candidiasis (Fig. 5) (Borghi et al,

2015). The same requirement of NLRC4 was observed in the case of *A. fumigatus* lung infection (Iannitti et al, 2016). In response to the infection, IL-22 cytokine production is induced and activates NLRC4, which then sustains the production of the IL-1 receptor antagonist IL-1Ra, which acts as a suppressor of inflammasome activity (Fig. 5) (Iannitti et al, 2016; Petrasek et al, 2012). Thus, negative control of NLRP3 activation by NLRC4/IL1Ra reduces inflammatory response, but allows host cells to control fungal infection. In contrast, when NLRC4 is absent, NLRP3 activation induces an exaggerated and ineffective inflammatory response with high levels of neutrophil recruitment. During *P. brasiliensis* infection, NLRC4 inflammasome is activated and inhibits NLRP3, decreasing the production of IL-1β and IL-18 (Souza et al, 2021). Similarly, in this model, mice lacking NLRC4 experienced resistance and lower fungal burden compared to WT animals. These discrepancies, i.e., activation or not of NLRC4 along with NLRP3, could be attributed to differences in the infection models, fungal strains, and microbiota composition. Whether NLRC4 is activated by bacteria that invade cells following mucosal barrier disruption caused by *Candida* species remains uncertain and warrants further investigation (Fig. 5). Indeed, dysbiosis is a well-known factor in the regulation of inflammasome and immune activation of mucosal organs (Watanabe et al, 2021; Manshouri et al, 2024; Xue et al, 2019).

The molecular mechanism by which NLRC4 inhibits NLRP3 inflammasome activity remains unknown, but clinical evidence confirms that a certain NLRC4 polymorphism is associated with increased risk of fungal colonization in cystic fibrosis patients and infection (Iannitti et al, 2016; Zhong et al, 2022a). Further studies need to be performed to understand the precise role of NLRC4 and how different inflammasomes can interact with each other to either induce a protective response or promote infection.

NLRC4 is traditionally recognized as a cytosolic sensor of bacterial flagellin and components of the bacterial type-III secretion system, with its activation often being mediated via NAIP (NLR family apoptosis inhibitory proteins) sensors that confer ligand specificity (Miao et al, 2010; Zhao et al, 2011) (Fig. 5). However, the mechanisms by which fungal pathogens, such as *C. albicans* and *A. fumigatus*, activate NLRC4 remain unclear. Currently available evidence indicates NLRC4 involvement in immune responses to these fungi, but direct fungal ligands have not been identified. It is speculated that NLRC4 may not recognize fungal PAMPs directly; instead, its activation could be indirectly modulated by bacterial dysbiosis resulting from fungal infection that introduces bacterial ligands known to activate NAIPs and NLRC4. Indeed, the role of NAIPs in fungal infections has not been sufficiently characterized, representing a critical gap in our understanding of how fungal infections influence inflammasome pathways traditionally associated with bacterial sensing. Further studies are warranted to determine whether NAIPs are capable of recognizing components derived from fungi, or if the activation of the NLRC4 inflammasome during fungal infection depends on secondary signals originating from bacterial cohabitants or host factors induced by the fungus.

## Inflammasome-dependent IL-1 signaling

Signaling through inflammasome-dependent IL-1 cytokines (IL-1β and IL-18) is protective in invasive fungal infections. During

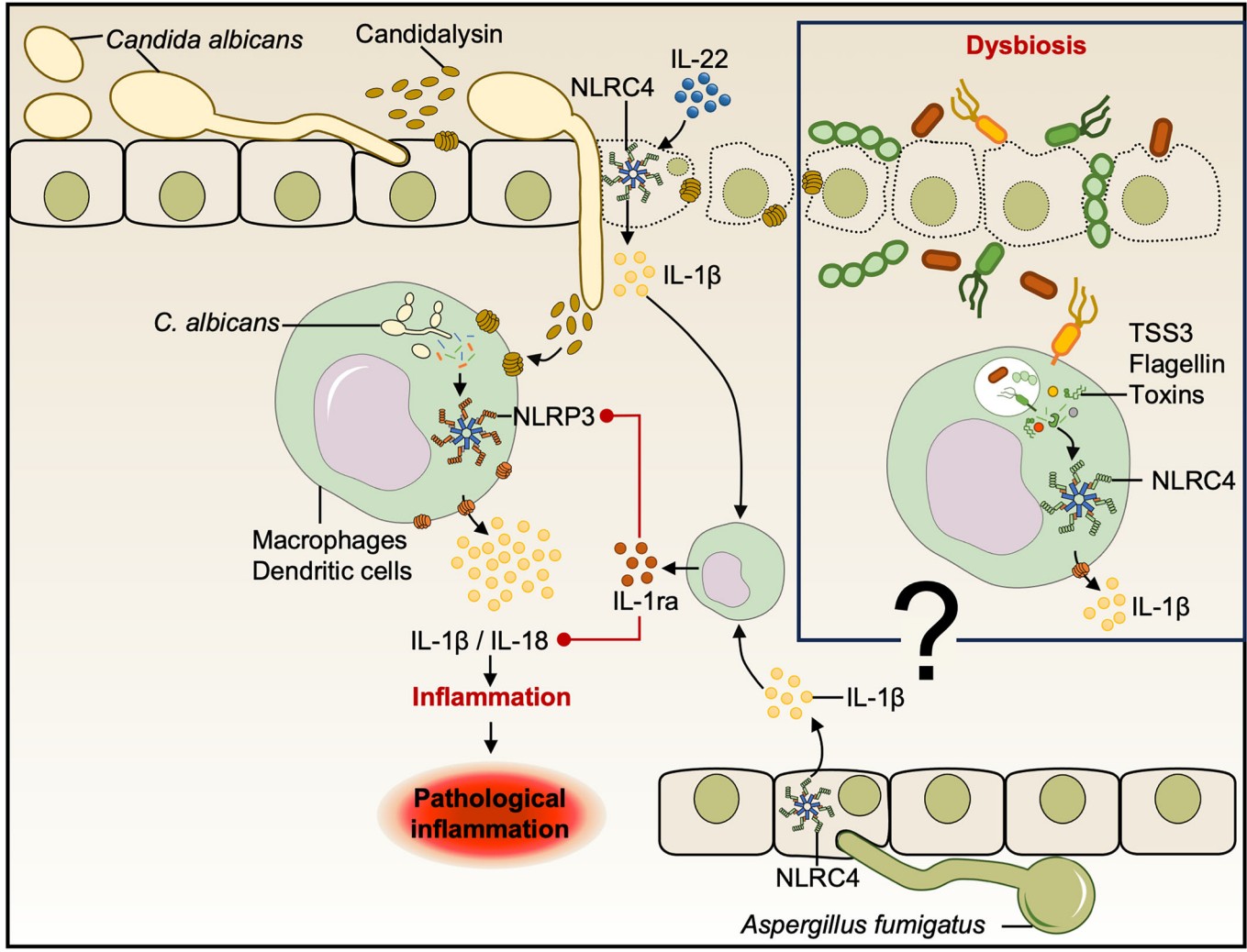

**Figure 5. NLRC4 inflammasome activation during fungal infection of mucosal barriers.**

*Candida albicans* can breach mucosal barriers and invade deeper tissues, facilitated by the pore-forming protein candidalysin. IL-22 induction promotes NLRC4 inflammasome activation, leading to the release of IL-1β and IL-1ra. Due to its antagonistic role with IL-1 receptors, IL-1ra inhibits the NLRP3 inflammasome (red dotted lines), preventing excessive proinflammatory cytokine production. In *C. albicans* infection, the protective role of NLRC4 is highlighted by the increased susceptibility of $Nlrc4^{-/-}$ mice. Similarly, NLRC4 confers protection during *A. fumigatus* lung colonization and infection. However, the ligand and mechanisms mediating NLRC4 activation in fungal infections remain unknown. In addition, one hypothesis is that bacterial dysbiosis induced by fungal infection of mucosal barriers may directly activate NLRC4.

pulmonary aspergillosis, $Il$-$1b^{-/-}$ mice exhibit significantly higher mortality rates compared to WT mice, whereas $Il$-$18^{-/-}$ mice show only partial susceptibility (Karki et al, 2015). Therefore, it would be of interest to assess the susceptibility of mice lacking both IL-1β and IL-18. Furthermore, loss of IL-1R increases susceptibility to invasive pulmonary aspergillosis (IPA) by impairing neutrophil recruitment and their survival (Ralph et al, 2021; Caffrey et al, 2015). Likewise, treatment with anakinra, an IL-1R antagonist, further confirms the role of IL-1 signaling in the host response to *A. fumigatus* infection (Gresnigt et al, 2016).

In terms of human polymorphisms, *IL1B* has been strongly associated with the development of IPA in patients with hematological disorders. Interestingly, while no significant association has been observed for individual polymorphisms in *IL1A* or *IL1RA* with IPA susceptibility, one was described with *IL1RA* and *IL1B* by the means of haplotype analyses (Sainz et al, 2008).

Similarly, during systemic candidiasis, IL-1 signaling is essential for controlling fungal invasion. Mice deficient in IL-1β or IL-18 production are more susceptible to systemic *C. albicans* infection. IL-1β appears to regulate immune cell recruitment, whereas IL-18 primarily controls IFN-γ production (Stuyt et al, 2002; Vonk et al, 2006; Netea et al, 2003a). Moreover, recent studies showed that IL-1R signaling in non-hematopoietic cells prevents fatal *Candida* dissemination by blocking a metabolomic shift during infection (Gresnigt et al, 2016).

Together, these data highlight the critical role of inflammasome-dependent IL-1 cytokine as a key mediator of antifungal protection.

## Other inflammasome regulators?

Another cytosolic sensor, the innate immune sensor Z-DNA binding-protein 1 (ZBP1), was recently demonstrated to be involved in NLRP3 inflammasome activation during *C. albicans* and *A. fumigatus* infection (Banoth et al, 2020; Briard et al, 2021a). ZBP1 has been most extensively characterized in the antiviral immunity context, where it interacts with other RIP homologous interaction motif domain (RHIM)-containing proteins through RHIM1 and RHIM2, thereby contributing to immune response transduction (Zhan et al, 2024). Evidence from studies investigating ZBP1 deficiency in macrophages supports its role in the activation of PANoptosis (Banoth et al, 2020), which is characterized by caspase-1 activation (pyroptosis), caspase-8, caspase-3, and caspase-7 triggering (apoptosis), and MLKL phosphorylation (necroptosis) (Chen et al, 2023b; Briard et al, 2021b). However, the precise activation mechanism of ZBP1 during fungal infection remains unclear; it may primarily function as a scaffold within the PANoptosome complex, acting independently of direct ligand binding. Moreover, the specific ligand(s) responsible for ZBP1 activation in this context remain unknown, necessitating further investigation. It is also unclear whether ZBP1 directly senses fungal Z-DNA or Z-RNA, or whether it could detect RNA derived from mycoviruses. Indeed, the function of ZBP1 is principally known from the context of viral infections (Zhan et al, 2024).

## Conclusions and future perspectives

Fungal pathogens are eukaryotic –mostly extracellular– organisms with a complex cell wall structure and metabolism. Historically, fungal infections have received less attention compared to other pathogens. However, modern societal and medical advances, combined with climate change, have substantially increased the pressure that fungi exert on human and animal health.

Elucidating the immune mechanisms involved in fungal diseases is challenging, but critical for understanding their pathophysiology and developing new therapeutic strategies. Indeed, the morphological plasticity of fungi contributes to this complexity, as multiple stages can obviously induce distinct and highly complex immune responses. For instance, dimorphic yeasts or conidia produced by molds are transition stages to subsequent hyphal or mycelial forms, with distinct cell wall composition and thus a diverse set of PAMPs. Consequently, hosts require very specialized defense mechanisms—which are greatly different from those against viruses or bacteria—to specifically restrict fungal growth and provide efficient protection.

Considering this point, our review highlights the pivotal role of cytosolic sensing in controlling fungal infections. Extracellular PRRs serve as the first line of defense, bridging surveillance between the host cell surface and cytosolic sensors. We described here in detail how fungal components can enter the cytoplasm, triggering type-I IFN signaling and inflammasome activation. Both can thereafter enhance antifungal immunity and, in some cases, exacerbate immunopathology and organ damage (Fernandes et al, 2018; Lachat et al, 2022).

Moreover, the discovery that fungal extracellular vesicles can release PAMPs into the host cytoplasm at locations distant from the initial infection site introduces an additional layer of complexity. Additionally, our review highlights the importance of cytosolic sensors that may not detect currently recognized fungal PAMPs or might respond to yet unidentified fungal signals. These observations draw attention to two critical factors in fungal immunopathology: the role of pathogen-induced ER stress in triggering immune responses, and the influence of the microbiota on immune regulation. Dysbiosis caused by invasive fungal pathogens may provoke an unconventional immune response, adding another layer of complexity to the host–pathogen interaction. Finally, in line with the growing appreciation of the roles of microbiota, mycoviruses infecting fungal pathogens undoubtedly shape a distinct host immune response, representing a promising avenue for future discoveries (Hitzler et al, 2025; Rocha et al, 2025). Recent studies have highlighted the significant impact of these viruses, such as the *A. fumigatus* dsRNA virus AfuPmV-1M, on fungal fitness and virulence. These findings reveal a complex interplay between fungal pathogens, their resident viruses, and host immunity, particularly involving cytosolic RNA-sensing pathways like MDA5/MAVS. Integrating mycovirus biology into fungal immunology promises new insights and therapeutic targets for invasive fungal infections (Rocha et al, 2025).

Many cytosolic sensors have already been shown to be essential during fungal infection. However, the roles of others such as the RIG-I, DDX/DHX RNA sensors, and the IFI16 DNA sensor remain incompletely understood.

Several emerging fungal pathogens have become increasingly important in recent years. It is therefore critical for the scientific community to also address immune recognition and responses to these new and concerning pathogens. The first example is *Candida auris*, which emerged in 2009 and has since spread worldwide, causing severe infections that are difficult to treat (Lionakis and Chowdhary, 2024). Recently, coccidioidomycosis, caused by *Coccidioides immitis* or *C. posadasii*, has emerged as a serious pulmonary fungal disease (Donovan et al, 2025; Galgiani and Kauffman, 2024). Similarly, mucormycoses, caused by mucormycetes such as *Rhizopus species*, have also become a significant global health concern, exemplified by outbreaks during the COVID-19 pandemic and the rise of antifungal resistance (Hoenigl et al, 2022; Steinbrink and Miceli, 2021). Notably, climate change and global warming are further expanding the geographic range and impact of these fungi by altering ecosystems and host susceptibility (Seidel et al, 2024). Despite their growing importance, our understanding of cytosolic immune-sensing mechanisms and host responses to these emerging pathogens remains limited. Addressing these knowledge gaps is critical for developing effective diagnostics and therapies, underscoring the need for future research to broaden the scope beyond well-studied fungi and anticipate new fungal threats in a rapidly changing world.

By exploring the aforementioned aspects (also summarized in Box 3), we will gain a deeper understanding of fungal pathogenesis and host immunity. Further research is essential to elucidate how fungal components access the cytosol and activate immune sensors under these various conditions, shedding light on both protective and pathological outcomes. This comprehensive perspective holds promise for guiding the development of innovative strategies to prevent and treat invasive fungal infections.

Box 3   Questions and future directions

- The cGAS–STING–autophagy axis in antifungal defense requires further study; especially how cGAS sensing of dsDNA activates STING to induce autophagy and regulate host defense, and how STING interacts with Src to modulate SYK signaling during *Candida* infection warrants further investigation.
- Pathogen-induced ER stress may act upstream of receptor activation; fungal infection-triggered ER stress could be sensed by NOD1/2, with IRE1α linking ER stress to innate immunity.
- The impact of mitochondrial perturbations on danger signaling is unclear: fungal infection triggers ROS and K$^+$ efflux for NLRP3 activation, but sources of cytosolic DNA and the role of mitochondrial ROS in AIM2/NLRP3 activation need clarification.
- More work is needed to define inflammasome-independent caspase functions in fungal infection; caspase-8 regulates IL-1β transcription and processing, and caspase-11 may drive pore formation and IL-1α release in a cell type-specific manner.
- The influence of microbiota on antifungal immunity and cytosolic sensing thresholds is an emerging area, especially as fungal pathogens can induce dysbiosis.
- The molecular mechanism of NLRC4 inflammasome activation in fungal infections remains unclear; fungal ligands have not been identified, and activation may be indirect via bacterial dysbiosis. The role of NAIPs in these models remains unexplored and requires further study.
- The impact of mycoviruses on host sensing and disease outcomes through RLR engagement has not been extensively studied and could be investigated experimentally.
- Human genetics should be leveraged to validate these pathways in infection; current human data for non-canonical caspase and AIM2 pathways are lacking, as most insight comes from murine models.
- Emerging pathogens such as *Candida auris*, *Coccidioides immitis*, *Coccidioides posadasii*, and mucormycetes remain understudied in the context of cytosolic immune sensing, despite their increasing incidence, highlighting the urgent need to investigate their host recognition and immune evasion strategies.

# Peer review information

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

## Acknowledgements

The authors acknowledge many investigators in the field whose primary data could not be cited in this review because of space limitations. The authors acknowledge Servier Medical Art for providing the graphical template. This work was supported by the patient association "*Vaincre la Mucoviscidose*" (VLM) and "*Association Gregory Lemarchal*": RF20230503265, RF20240503504; and "*Agence National de la Recherche*" (ANR): ANR-23-CE15-0013-03, ANR-24-CE15-2995-03 to BB.

## Author contributions

**Sandra Khau**: Conceptualization; Writing—original draft; Writing—review and editing. **Guillaume Desoubeaux**: Writing—original draft; Writing—review and editing. **Mustapha Si-Tahar**: Writing—original draft; Writing—review and editing. **Elise Biquand**: Writing—original draft; Writing—review and editing. **Benoit Briard**: Conceptualization; Supervision; Funding acquisition; Validation; Writing—original draft; Writing—review and editing.

## Disclosure and competing interests statement

The authors declare no competing interests. GD is the president of the board of the French Society of Medical Mycology.

