## [Peer Review File · The EMBO Journal]

Cytosolic receptors and signaling in antifungal immunity

Sandra Khau, Guillaume Desoubieux, Mustapha SI-TAHAR, Elise Biquand, and Benoit Briard

Corresponding author(s): Benoit Briard (benoit.briard@inserm.fr)

Review Timeline:

Submission Date:	2nd Apr 25
Editorial Decision:	25th Apr 25
Revision Received:	18th Sep 25
Accepted:	28th Sep 25

Editor: Ioannis Papaioannou

Transaction Report:

Dear Benoit,

Thank you again for submitting your Review manuscript (EMBOJ-2025-120960-T) to The EMBO Journal for our consideration, and for your patience during peer review. Your manuscript has now been seen by three experts in the field, and we have received their detailed and constructive comments, which are included below.

I am pleased to say that, as you will see, the referees are all very supportive and recognize that your Review article is of high general interest, novel, and interesting. They also identify, however, a few limitations and make several constructive suggestions for the improvement of the manuscript and the figures. Among other suggestions, they point out specific topics that could be included in the Review for better contextualization of the role of intracellular sensing in antifungal immunity, including human genetic evidence linking sensing pathways to fungal infection susceptibility, and data from other pathogens. They also suggest streamlining some sections of the manuscript for better readability, and they also list specific corrections that should be made.

We largely agree with the referees' points and advice, and we think that your Review article and its impact on the field will benefit from the suggested changes and additions. In light of this input from the referees, I would like to invite you to revise your manuscript taking the referees' suggestions on board, and submit the revised version to our online manuscript tracking system. Please include in your resubmission a point-by-point response to the referees' comments addressing all suggestions and concerns.

While revising your text, we would kindly also request you to:

- add the heading "Abstract" before the summary on the first page of the manuscript
- provide a list of up to 5 relevant keywords after the Abstract
- make sure that the headings (and subheadings) of the main text sections are presented consistently (e.g., regarding numbering); please do not use colored fonts, instead use bold or italic fonts for different levels
- add a "Disclosure and competing interests statement" before the References list.

I would also like to mention that you can use (up to 4-5) Text Boxes for information that is not central or essential in the main text for the logical flow of the manuscript, but still relevant to the topic of the Review article, or providing more detailed information on particular sections without distracting from the main message; please see also the related point 1 of referee #2. Please remember that the numbered Text Boxes must be called out where appropriate in the main text (e.g. "Text Box 1", "Text Box 2" etc.), and each one should have a brief heading.

I would strongly recommend including an "In need of answers" or "Questions and future directions" Text Box at the end of the main text summarizing important questions in the field that you think should be prioritized in the future, along with brief suggestions on what needs to be done to address these questions. This Text Box would be complementary to the "Conclusions and Future Perspectives" section of the manuscript.

Please include the numbered Text Boxes at the end of the main text, before the list of References.

Regarding the Figures, we think that they are helpful for visually supporting the text and summarizing the discussed mechanisms, and that their design is adequate and professional. Please see the related points of the referees for the improvement of the Figures. I would also like to kindly ask you to make sure during revision that all Figures are scientifically accurate and that all of their graphical elements are clearly defined/explained in detail in the respective legends. If there are certain aspects of your Figures that are based upon assumptions or where the scientific data remain ambiguous (for example, schematically depicting a presumed direct protein-protein interaction, protein shape or subcellular localization etc.), please explain in the respective legends or add a comment so that we can work with you on an accurate depiction. Please ensure that the directionality and nature of interactions is presented accurately.

Please also note that:

- If the Figures or single panels of the Figures have been adapted from a published Figure, this information must be added to the Figure legend (e.g., 'Adapted from...' or 'Based on...'). I will then discuss with you if a reference and permission will be necessary. Please only re-use Figures or parts of a Figure if this is essential for understanding the concept communicated. Often a reference to a previous paper will suffice.
- If the Figures contain re-used images or elements of images, including schematics, micrographs or photos, please make sure that you have the permission/license to publish them (this also applies to your own previous work, if the journal you previously published in retains copyright). Certain "creative commons" open access licenses, such as CC-BY 4.0, allow re-use without additional formal permissions. All re-used material must be explicitly cited.
- If you use an image database for scientific iconography (such as BioRender), please acknowledge it in the respective Figure legends and make sure that you have a license that allows for publication in an academic journal.

Please also note that as part of the EMBO publications' Transparent Editorial Process, The EMBO Journal publishes online a Peer Review File along with each accepted manuscript. This File will be published in conjunction with your review article and will include the referee reports, your point-by-point response, and all pertinent correspondence relating to the manuscript. You can opt out of this by letting the editorial office know (contact@embojournal.org). If you do opt out, the Peer Review File link will point to the following statement: "No Peer Review File is available with this article, as the authors have chosen not to make the review process public in this case."

We look forward to your revised Review article addressing the above points as soon as possible. Please let us know if you have any questions or comments you would like to discuss with me. When you are ready to re-submit your revision, please use the link:

<https://emboj.msubmit.net/cgi-bin/main.plex>.

Best regards,

Ioannis

Referee #1:

This review is significant as it synthesizes current knowledge on the multifaceted role of innate immune receptors, particularly focusing on the cytosolic sensors that detect fungal pathogens—a critical aspect given that fungi can evade traditional extracellular detection mechanisms. It highlights how pattern recognition receptors (PRRs), including membrane-bound Toll-like receptors (TLRs) and C-type lectin receptors (CLRs), initiate tailored immune responses by binding to distinct pathogen-associated (PAMPs) and damage-associated molecular patterns (DAMPs). Importantly, the study extends this understanding by elucidating the roles of cytosolic receptors such as Nod-like receptors (NLRs), AIM2-like receptors (ALRs), and RIG-I-like receptors (RLRs), which become crucial when phagocytosed fungi escape into the cytoplasm. Overall, the review underscores the complexity and specificity of host-pathogen interactions, suggesting that a deeper comprehension of these signaling pathways could inform novel therapeutic strategies for managing fungal infections.

Major concerns:

- 1) Although several intracellular fungal sensing mechanisms have been identified in recent years, many of them still require further validation, and there is a lack of supporting human genetic evidence. For example, genetic mutations linked to fungal susceptibility—such as those in Dectin-1, CARD9, STAT3, IL-17R, IL-17A, and ACT1—are frequently observed in patients with chronic mucocutaneous candidiasis. These findings strongly support the critical role of Dectin signaling, STAT3 signaling, and IL-17 signaling in antifungal immunity. I suggest the authors review the literature and incorporate genetic evidence linking intracellular sensing pathways to fungal infection susceptibility, as this would further strengthen the significance of these pathways.
- 2) Although this review focuses on intracellular fungal sensing, I recommend that the authors briefly summarize well-established antifungal signaling pathways, such as Dectin signaling in monocytes/macrophages, STAT3 signaling in Th17 cells, and IL-17/ACT1 signaling in epithelial cells. Including these pathways would provide a more comprehensive overview of antifungal immunity and help contextualize the role of intracellular sensing.

Minor concerns that should be addressed:

- 1) The activation mechanism of ZBP1 remains speculative. It would help to more clearly state that it's unknown whether ZBP1 directly senses fungal Z-DNA/RNA or viral elements from mycoviruses.
- 2) There are occasional minor grammatical issues (e.g., "plays an essential role" → "play an essential role" in the abstract). A round of copyediting for grammar and flow would be helpful.
- 3) Use consistent formatting for gene and protein names (e.g., IFIH1 vs. MDA5; gene names in italics, proteins in uppercase regular font, if following journal guidelines).
- 4) Some promising future directions like the role of autophagy, ER stress in receptor activation, mitochondrial damage, and inflammasome-independent roles of caspases are touched upon but could be more explicitly outlined in the "Conclusion and Future Perspectives" section.

Referee #2:

In this review, Khau and colleagues explore the role of cytosolic sensors in the immune response to fungi. The review is organized into sections, each focusing on a specific sensor class or family. The topic is both timely and underexplored in the current literature. The authors provide a comprehensive overview of cytosolic receptors, their functions, and potential mechanisms underlying their involvement in disease, drawing from studies on both mouse models and humans. However, this reviewer feels there are still areas for improvement, especially considering the journal's broad readership.

1. At certain points, the level of detail in describing specific studies detracts from the main message. Streamlining these sections could improve clarity and make the review more reader-friendly. Moreover, highlighting key aspects of the discussion in the context of available literature on other pathogens could broaden the review's appeal and engage a wider readership.
2. The figures provide a valuable aid in illustrating the complex mechanistic details and supporting the text. However, I would suggest removing the implications of the different mechanisms on mouse survival shown at the bottom of each figure. This inclusion can be misleading for two reasons: (i) some studies present conflicting findings, making general conclusions premature; and (ii) overemphasizing mouse-specific mechanisms may undervalue the broader relevance of these molecules and pathways to human disease.

Minor points:

3. The incidence rates of invasive fungal infections are estimates and should be clearly defined as such in the introduction.
4. In the section describing the role of NOD2 in Aspergillosis, it may be worthwhile to include a statement on the relevance of well-studied gene variants, typically associated with inflammatory diseases, as resistance factors to aspergillosis in humans (as the authors have done for other of the receptors discussed).
5. The conflicting results regarding the role of GAG should be presented in a more balanced manner. While the contribution of GAG to inflammasome activation is well-detailed, the mechanisms proposed in studies suggesting that GAG might block IL-1 β secretion would benefit from further elaboration.
6. In the figures, the full names of Dectin-1 and Dectin-2 should be displayed for clarity.

Referee #3:

This review article "Cytosolic Sensing Mechanisms in Antifungal Immunity: Unraveling Their Critical Roles" by Briard and colleagues provides a well-structured and balanced overview of the cytosolic pattern-recognition receptors in anti-fungal immunity. The authors clearly discussed the most exciting development in the field relating to the roles of NOD1/NOD2, RIG-I-like receptors, cGAS and STING, and inflammasome signaling. Given that most of the literature are on two fungal pathogens, *Candida albicans* and *Aspergillus fumigatus*, the focus of this review is on the discussion of these two pathogens, which seems appropriate. This review is very timely and will appeal to the audience in the field. I have made several suggestions below, which I feel would increase the scientific appeal and accuracy of this excellent review article.

1. In the Introduction section, I suggest expanding the scope a little bit more between lines 70-74. Here the authors argue that cytosolic sensors have two functional categories represented by interferon signaling and inflammasome activation. As discussed later in this review, NOD1, NOD2, RIG-I-like receptors, cGAS and STING can also trigger the activation of NF- κ B pathways. Therefore, the scope should be expanded here to include three categories of the functionalities for these cytosolic sensors.
2. To provide a bit more context in the section starting from lines 277-295, it might be helpful to mention whether IL-1 and/or IL-18 signalling is protective against fungal infection, which would provide a segue to talk about each inflammasome. Also, it would be beneficial to add an extra level of insights here on whether there are unique contributions comparing IL-1/IL-18 axes versus pyroptosis during fungal infections.
3. In the section on the GAG of *A. fumigatus* (line 368), the authors mentioned another study showing GAG in reducing IL-1 β secretion. A citation to that work is missing.
4. Since NLRC4 is a sensor of bacterial flagellin and appears to have a role against *C. albicans* and *A. fumigatus*, please can the authors add a bit more insights as to how fungal pathogens activate NLRC4? For example, do they speculate it is a ligand interaction or not? Perhaps also mention on the role of NAIPs or missing knowledge gaps on NAIPs in these models?
5. The discussion on interferon-inducible GTPases in driving inflammasome activation during *A. fumigatus* (lines 421-441): In addition to IRGB10, can the authors expand this discussion to include the roles or lack of a role for GBPs that often synergise

with IRGB10?

6. In the Conclusion and Future Perspective section, I suggest adding the authors' thoughts on the knowledge gaps on other less studied and potentially emerging fungal pathogens that researchers should be looking out for.
7. References #42 and #43 are duplicated.
8. Figure 1: What are the brown lines emanating from NOD1 and NOD2? Also, I suggest labelling the conidia spores if these green circles are the authors are referring to.
9. Figure 5: "Flagelline" should be "Flagellin", and "Toxines" should be "Toxins".

Dear Dr. Papaioannou,

We are very grateful for your continued support and express our sincere thanks to the reviewers for stating that our review is "well-structured and balanced overview of the cytosolic pattern-recognition receptors in anti-fungal immunity" and "is significant as it synthesizes current knowledge on the multifaceted role of innate immune receptors"

Attached below are our responses to the suggestions by the reviewers'.

Reviewers' Comments:

Referee #1:

This review is significant as it synthesizes current knowledge on the multifaceted role of innate immune receptors, particularly focusing on the cytosolic sensors that detect fungal pathogens—a critical aspect given that fungi can evade traditional extracellular detection mechanisms. It highlights how pattern recognition receptors (PRRs), including membrane-bound Toll-like receptors (TLRs) and C-type lectin receptors (CLRs), initiate tailored immune responses by binding to distinct pathogen-associated (PAMPs) and damage-associated molecular patterns (DAMPs). Importantly, the study extends this understanding by elucidating the roles of cytosolic receptors such as Nod-like receptors (NLRs), AIM2-like receptors (ALRs), and RIG-I-like receptors (RLRs), which become crucial when phagocytosed fungi escape into the cytoplasm. Overall, the review underscores the complexity and specificity of host-pathogen interactions, suggesting that a deeper comprehension of these signaling pathways could inform novel therapeutic strategies for managing fungal infections.

We appreciate the Referee #1 for noting that our review is significant as is described "the complexity of the innate immune receptors and particularly highlighting the role of the cytosolic sensors for extracellular pathogens". We have followed the recommendations of this reviewer in the revised manuscript.

Major concerns:

1) Although several intracellular fungal sensing mechanisms have been identified in recent years, many of them still require further validation, and there is a lack of supporting human genetic evidence. For example, genetic mutations linked to fungal susceptibility—such as those in Dectin-1, CARD9, STAT3, IL-17R, IL-17A, and ACT1—are frequently observed in patients with chronic mucocutaneous candidiasis. These findings strongly support the critical role of Dectin signaling, STAT3 signaling, and IL-17 signaling in antifungal immunity. I suggest the authors review the literature and incorporate genetic evidence linking intracellular sensing pathways to fungal infection susceptibility, as this would further strengthen the significance of these pathways.

We have incorporate the genetic evidence linking intracellular sensing pathways to fungal infection susceptibility in the main text.

NOD2 (line 135 – 139): Human genetic studies indicate that NOD 2 mutations modulate susceptibility to fungal infections such as invasive aspergillosis by altering cytokine responses and phagocytosis, with some loss-of-function variants paradoxically conferring resistance, underscoring its complex role in antifungal immunity³³.

IFIH1 – MAVS (line 185 – 188): "The critical role of the MDA5/MAVS pathway was confirmed by the identification of IFIH1 and MAVS polymorphisms, which are associated with an increased incidence of invasive infection in hematopoietic stem-cell transplantation (HSCT) recipients⁵⁹. Indeed, patients carrying IFIH1 mutations show altered pro- and anti-inflammatory cytokine profiles, correlating with increased infections risk⁵⁰."

NLRP3 (line 365 – 369): “In terms of human genetic evidence, several polymorphisms in the *Nlrp3* gene have been shown to influence susceptibility and severity of fungal infections by modulating inflammasome activation and IL-1 β production. Gain-of-function variants of *Nlrp3* can result in heightened inflammasome activation, contributing to both protective immunity and excessive inflammation, highlighting NLRP3’s dual role in antifungal defense and inflammation^{114,115}.”

NLRC4 (Line 473 – 475): “The molecular mechanism by which NLRC4 inhibits NLRP3 inflammasome activity remains unknown, but clinical evidence confirms that a certain NLRC4 polymorphism is associated with risk of fungal colonization in cystic fibrosis patients and fungal infection^{144,149}.”

IL1B (line 508 – 507): “In terms of human polymorphisms, IL1B have been strongly associated with the development of IPA in patients with hematological disorders. Interestingly, while no significant association has been observed for individual polymorphisms in IL1A or IL1RA with IPA susceptibility, one was described with IL1RA and IL1B by the means of haplotype analyses¹⁵⁵.”

2) Although this review focuses on intracellular fungal sensing, I recommend that the authors briefly summarize well-established antifungal signaling pathways, such as Dectin signaling in monocytes/macrophages, STAT3 signaling in Th17 cells, and IL-17/ACT1 signaling in epithelial cells. Including these pathways would provide a more comprehensive overview of antifungal immunity and help contextualize the role of intracellular sensing.

We acknowledge Referee #1’s comment and have now added a paragraph to the introduction (lines 55–66, see below) highlighting the importance of the C-type lectin pathway, STAT1, STAT3, and the IL-17/Th17 axis.

“Several immune signaling pathways have been demonstrated to play pivotal roles in human host defense against fungal pathogens. Mutations affecting the C-type lectin receptor (CLR) pathway, including those in *CLEC7A* (encoding Dectin-1) and *CARD9*, are strongly associated with increased susceptibility to chronic mucocutaneous or vulvovaginal candidiasis as well as fungal meningitis^{7–15}. In parallel, the JAK–STAT signaling axis is critical for orchestrating antifungal immunity, primarily through the regulation of effective Th17 responses. Genetic mutations in STAT1 or STAT3 similarly predispose individuals to recurrent mucocutaneous candidiasis, fungal meningitis, and invasive pulmonary fungal infections^{16–18}. Consistent with this, human genetic defects directly impacting the IL-17 pathway confer comparable susceptibility to these fungal pathologies^{19–23}. Collectively, these observations highlight the central role of coordinated innate and adaptive immune responses in antifungal host defense.”

Minor concerns that should be addressed:

1) The activation mechanism of ZBP1 remains speculative. It would help to more clearly state that it’s unknown whether ZBP1 directly senses fungal Z-DNA/RNA or viral elements from mycoviruses.

We agree and have revised the sentence to clearly state that it remains unknown whether ZBP1 directly senses fungal Z-DNA/RNA or viral elements derived from mycoviruses.

Line 526 – 529: “However, the precise activation mechanism of ZBP1 during fungal infection remains unclear; it may primarily function as a scaffold within the PANoptosome complex, acting independently of direct ligand binding. Moreover, the specific ligand(s) responsible for ZBP1 activation in this context remain unknown, necessitating further investigation.”

2) There are occasional minor grammatical issues (e.g., "plays an essential role" → "play an essential role" in the abstract). A round of copyediting for grammar and flow would be helpful.
We have carefully reviewed the text and corrected the grammatical errors

3) Use consistent formatting for gene and protein names (e.g., IFIH1 vs. MDA5; gene names in italics, proteins in uppercase regular font, if following journal guidelines).
We apologize for this mistake we have update the text.

4) Some promising future directions like the role of autophagy, ER stress in receptor activation, mitochondrial damage, and inflammasome-independent roles of caspases are touched upon but could be more explicitly outlined in the "Conclusion and Future Perspectives" section.
We agree and we have now developed our Conclusion and Future Perspectives section and added a Text Box Questions and future directions.

Referee #2:

In this review, Khau and colleagues explore the role of cytosolic sensors in the immune response to fungi. The review is organized into sections, each focusing on a specific sensor class or family. The topic is both timely and underexplored in the current literature. The authors provide a comprehensive overview of cytosolic receptors, their functions, and potential mechanisms underlying their involvement in disease, drawing from studies on both mouse models and humans. However, this reviewer feels there are still areas for improvement, especially considering the journal's broad readership.

We are grateful to Referee #2 for the insightful review and constructive comments, which have helped us to improve the manuscript and broaden its accessibility

1. At certain points, the level of detail in describing specific studies detracts from the main message. Streamlining these sections could improve clarity and make the review more reader-friendly. Moreover, highlighting key aspects of the discussion in the context of available literature on other pathogens could broaden the review's appeal and engage a wider readership.

We understand the point raised by Referee #2 and have attempted to remove sections where the level of detail was excessive. As suggested by the editor, we also added two text boxes. However, we may have overlooked some parts of the manuscript. If so, please let us know the specific sections, and we will update them accordingly.

2. The figures provide a valuable aid in illustrating the complex mechanistic details and supporting the text. However, I would suggest removing the implications of the different mechanisms on mouse survival shown at the bottom of each figure. This inclusion can be misleading for two reasons: (i) some studies present conflicting findings, making general conclusions premature; and (ii) overemphasizing mouse-specific mechanisms may undervalue the broader relevance of these molecules and pathways to human disease.

We understand Referee #2's point. We also hesitated to keep this graphical representation during our design. Following the recommendation, we have removed the mice survival graphical representation. Instead, we added the term "Pathological inflammation" in Figures 3 and 5.

Minor points:

3. The incidence rates of invasive fungal infections are estimates and should be clearly defined as such in the introduction.

We have revised our initial sentence to emphasize that the rates of invasive fungal infections are estimates, so as not to mislead the reader into thinking these numbers are confirmed.

4. In the section describing the role of NOD2 in Aspergillosis, it may be worthwhile to include a statement on the relevance of well-studied gene variants, typically associated with inflammatory diseases, as resistance factors to aspergillosis in humans (as the authors have done for other of the receptors discussed).

We have incorporate the human genetic evidence linking NOD2 pathway to fungal infection susceptibility in the main text.

NOD2 (line 135 – 139): “Human genetic studies indicate that NOD2 mutations modulate susceptibility to fungal infections such as invasive aspergillosis by altering cytokine responses and phagocytosis, with some loss-of-function variants paradoxically conferring resistance, underscoring its complex role in antifungal immunity⁴⁰.”

5. The conflicting results regarding the role of GAG should be presented in a more balanced manner. While the contribution of GAG to inflammasome activation is well-detailed, the mechanisms proposed in studies suggesting that GAG might block IL-1 β secretion would benefit from further elaboration.

We understand Reviewer 2’s point. However, based on the literature, we do not believe there are truly conflicting results regarding the effect of GAG on the inflammasome. Some studies indicate that GAG can activate the inflammasome in macrophages, whereas one study, using an *A. nidulans* strain, reported reduced IL-1 β release in PBMCs. Another study showed that GAG can induce IL-1Ra secretion in PBMCs, which may in turn reduce IL-1 β bioactivity.

The discrepancies likely arise from differences in the cell types used across studies. For example, LPS can activate the inflammasome without second activation in human monocytes but not in macrophages.

We have incorporated this aspect into the manuscript (line 351 – 369) to provide a more complete perspective on the role of GAG in inflammasome regulation.

6. In the figures, the full names of Dectin-1 and Dectin-2 should be displayed for clarity.

We have updated Figures 1 and 4 to clearly write the full names of Dectin-1 and Dectin-2.

Referee #3:

This review article "Cytosolic Sensing Mechanisms in Antifungal Immunity: Unraveling Their Critical Roles" by Briard and colleagues provides a well-structured and balanced overview of the cytosolic pattern-recognition receptors in anti-fungal immunity. The authors clearly discussed the most exciting development in the field relating to the roles of NOD1/NOD2, RIG-I-like receptors, cGAS and STING, and inflammasome signaling. Given that most of the literature are on two fungal pathogens, *Candida albicans* and *Aspergillus fumigatus*, the focus of this review is on the discussion of these two pathogens, which seems appropriate. This review is very timely and will appeal to the audience in the field. I have made several suggestions below, which I feel would increase the scientific appeal and accuracy of this excellent review article.

We thank the referee #3 for his thoughtful comments and constructive feedback on our manuscript.

1. In the Introduction section, I suggest expanding the scope a little bit more between lines 70-74. Here the authors argue that cytosolic sensors have two functional categories represented by interferon signaling and inflammasome activation. As discussed later in this review, NOD1,

NOD2, RIG-I-like receptors, cGAS and STING can also trigger the activation of NF- κ B pathways. Therefore, the scope should be expanded here to include three categories of the functionalities for these cytosolic sensors.

We thank Referee#3 for this comment. We have updated the text to indicate that the NF- κ B pathway is also regulated by cytosolic sensors.

Line (81 – 84): “*The cytosolic sensors can be primarily divided into three functional categories: those mediating interferon (IFN) signaling, those triggering the NF- κ B pathway and those mediating inflammasome activation.*”

2. To provide a bit more context in the section starting from lines 277-295, it might be helpful to mention whether IL-1 and/or IL-18 signalling is protective against fungal infection, which would provide a segue to talk about each inflammasome. Also, it would be beneficial to add an extra level of insights here on whether there are unique contributions comparing IL-1/IL-18 axes versus pyroptosis during fungal infections.

Indeed, we had omitted to add the evidence on the role of IL-1 signaling itself in controlling fungal infection. We have now updated the text by adding a section (line 502 – 517, see below) entitled “Inflammasome-dependent IL-1 signaling.

“Signaling through inflammasome-dependent IL-1 cytokines (IL-1 β and IL-18) is protective in invasive fungal infections. During pulmonary aspergillosis, *Il-1b*^{-/-} mice exhibit significantly higher mortality compared with WT mice, whereas *Il-18*^{-/-} mice show only partial susceptibility⁹⁴. Therefore, it would be of interest to assess the susceptibility of mice lacking both IL-1 β and IL-18. Consistently, loss of IL-1R increases susceptibility to invasive pulmonary aspergillosis (IPA) by impairing neutrophil recruitment and their survival^{150,151}. Likewise, treatment with anakinra, an IL-1R antagonist, further confirms the role of IL-1 signaling in the host response to *A. fumigatus* infection¹⁵².

In terms of human polymorphisms, *IL1B* have been strongly associated with the development of IPA in patients with hematological disorders. Interestingly, while no significant association has been observed for individual polymorphisms in *IL1A* or *IL1RA* with IPA susceptibility, one was described with *IL1RA* and *IL1B* by the means of haplotype analyses¹⁵⁵.

Similarly, during systemic candidiasis, IL-1 signaling is essential for controlling fungal invasion. Mice deficient in IL-1 β or IL-18 are more susceptible to systemic *C. albicans* infection. IL-1 β appears to regulate immune cell recruitment, whereas IL-18 primarily controls IFN- γ production^{78,153,154}. Moreover, recent findings showed that IL-1R signaling in non-hematopoietic cells prevents fatal *Candida* dissemination by blocking a metabolomic shift during infection¹⁵².

Together, these data highlight the critical role of inflammasome-dependent IL-1 cytokines as a key mediators of antifungal protection.“

3. In the section on the GAG of *A. fumigatus* (line 368), the authors mentioned another study showing GAG in reducing IL-1b secretion. A citation to that work is missing.

We have now included the reference and also update the text to follow the recommendation of the referee#2.

4. Since NLRC4 is a sensor of bacterial flagellin and appears to have a role against *C. albicans* and *A. fumigatus*, please can the authors add a bit more insights as to how fungal pathogens activate NLRC4? For example, do they speculate it is a ligand interaction or not? Perhaps also mention on the role of NAIPs or missing knowledge gaps on NAIPs in these models?

We have updated the manuscript to highlight the gap in our understanding of NLRC4 inflammasome activation and the role of NAIPs in recognizing fungal PAMPs. This update has been made between lines 479 and 492 (see below).

“NLRC4 is classically recognized as a cytosolic sensor of bacterial flagellin and components of the bacterial type III secretion system, with activation often mediated via NAIP (NLR family apoptosis inhibitory proteins) sensors that confer ligand specificity^{140,141} (Fig. 5). However, the mechanisms by which fungal pathogens such as *C. albicans* and *A. fumigatus* activate NLRC4 remain unclear. Current evidence shows NLRC4 involvement in immune responses to these fungi but does not identify direct fungal ligands. It is speculated that NLRC4 may not recognize fungal PAMPs directly, instead, its activation could be indirectly modulated by bacterial dysbiosis resulting from fungal infection that introduces bacterial ligands known to activate NAIPs and NLRC4. Indeed, the role of NAIPs in fungal infections has not been well characterized, representing a critical gap in understanding how fungal infections influence inflammasome pathways traditionally associated with bacterial sensing. Further studies are warranted to determine whether NAIPs are capable of recognizing components derived from fungi, or if the activation of the NLRC4 inflammasome during fungal infection depends on secondary signals originating from bacterial cohabitants or host factors induced by the fungus.”

5. The discussion on interferon-inducible GTPases in driving inflammasome activation during *A. fumigatus* (lines 421-441): In addition to IRGB10, can the authors expand this discussion to include the roles or lack of a role for GBPs that often synergise with IRGB10?

We have extended the discussion (Line 424 – 429) to address the differences observed between bacterial and fungal infections, highlighting that GBPs are required during bacterial infection but dispensable during *Aspergillus* infection.

6. In the Conclusion and Future Perspective section, I suggest adding the authors' thoughts on the knowledge gaps on other less studied and potentially emerging fungal pathogens that researchers should be looking out for.

We agree with the reviewer. In our initial submission, we focused on the host side; however, incorporating the perspective on fungal pathogens provides valuable future directions. Accordingly, we have updated the manuscript and highlighted three fungal pathogens that we believe should be the focus of future studies: the emerging *Candida auris*, coccidioidomycosis caused by *Coccidioides immitis* or *C. posadasii*, and mucormycosis, which is primarily caused by mucormycetes.

7. References #42 and #43 are duplicated.

We thank Referee #3 for this observation. We have updated our references.

8. Figure 1: What are the brown lines emanating from NOD1 and NOD2? Also, I suggest labelling the conidia spores if these green circles are the authors are referring to.

We apologize for the misunderstanding regarding Figure 1. We have reorganized the brown lines and added arrows to indicate that the gray elements originate from *A. fumigatus* and then bind to NOD2. We have also clearly labeled these elements as chitin. We have also clarify the green circles as resting conidia of *A. fumigatus* and then the germinated conidia/hyphae for the second form of *A. fumigatus*.

9. Figure 5: "Flagelline" should be "Flagellin", and "Toxines" should be "Toxins".

The update have been made on the Figure 5

Dear Benoit,

Congratulations on an excellent Review article! I am very pleased to inform you that it has been accepted for publication in The EMBO Journal. Thank you for comprehensively addressing the initially raised referee criticisms and the editorial requests for corrections and changes.

Your manuscript will be processed for publication by EMBO Press. It will be copy edited and you will receive page proofs prior to publication.

If you have any questions, please do not hesitate to contact the Editorial Office. Thank you for your contribution to The EMBO Journal. Working with you has been a pleasure!

Best regards,

Ioannis
